# Subslab ultra low velocity anomaly uncovered by and facilitating the largest deep earthquake

Weiwen Chen[1,2], Shengji Wei [1,3,4] ✉ & Weitao Wang [2] ✉

It is enigmatic that M8+ earthquakes can take place at depth greater than 600 km inside the slab, where the P-T conditions generally do not favor seismic slip rate (~m/s) on faults. Here we provide fresh insights to the initial rupture and mechanism of the Mw 8.3 Sea of Okhotsk earthquake by analyzing high-frequency (up to 0.8 Hz) teleseismic array data. We determine the relative location and timing of two early subevents, and the geometry and velocity perturbation of a nearby structure anomaly. We found a small-scale (~30 × 60 × 60 km) ultralow (−18 ± 2%) P-wave velocity anomaly located beneath the Pacific slab around the 660 km discontinuity. The volatile-bearing highly melted nature of the anomaly provides significant buoyancy, stressing the slab dramatically closer to the critical condition for thermal runaway weakening that allows the rupture to propagate beyond the metastable olivine wedge, forming M8+ events. Enormous velocity reduction urges for further mineral physics and geodynamic investigations.

Understanding the mechanisms behind deep earthquakes (with depths exceeding 300 km) is a complex task that requires further investigation. One appealing hypothesis is the occurrence of transformational faulting from olivine to spinel within the metastable olivine wedge (MOW), which offers a widely accepted explanation for the abrupt termination of deep seismicity below the 660 km discontinuity (660-D)[1,2]. However, it is challenging to comprehend how a confined area such as the MOW, which rapidly decreases in thickness with depth[1], can accommodate a magnitude 8 earthquake, unless the rupture occurs along a narrowly elongated fault plane aligned with the slab's strike or is facilitated by strong dynamic weakening, enabling the rupture to propagate beyond the MOW. Hence, the hypothesis of thermal runaway insatiability (TRI) emerges as a crucial additional mechanism to explain the occurrence of these rare large deep earthquakes, encompassing what is referred to as the "two-stage hypothesis" with transformational faulting as the initial stage and TRI dominates the following stage of the rupture[3–6]. To trigger such instability, a key requirement is the accumulation of high-strain/stress within the slab[7,8]. Nevertheless, the identification of the structure that hosts the extreme stress conditions necessary for triggering thermal runaway, as well as the process by which a small event within the MOW can evolve into a large earthquake, still remains under investigation[8,9].

Resolving the velocity structure and earthquake rupture process with higher resolution is essential for unraveling the concealed mechanisms of deep-source earthquakes. Advancing this understanding necessitates the application of waveform inversion/modeling at higher frequencies to image velocity structure and earthquake source processes more accurately. In recent studies, high-frequency seismic waveform modeling has proven valuable in constraining deep velocity structure anomalies with improved precision[10–12]. Notably, it has been commonly observed that waveform modeling requires velocity perturbations 1.5 to 2 times higher than those in travel time tomography models, resulting in sharper delineation of velocity anomalies that appeared blurred in the tomography models[10,13–15]. Capitalizing on their complementary characteristics, high-frequency (e.g., ~1 Hz) waveform modeling is often integrated with tomography results to obtain a more comprehensive understanding of subsurface structures[14,16].

[1]Earth Observatory of Singapore, Nanyang Technological University, Singapore, Singapore. [2]Institute of Geophysics, China Earthquake Administration, Beijing, China. [3]Asian School of the Environment, Nanyang Technological University, Singapore, Singapore. [4]Institute of Geology and Geophysics, Chinese Academy of Sciences, Beijing, China. ✉e-mail: shjwei@ntu.edu.sg; wangwt@cea-igp.ac.cn

In this study, we employ a waveform modeling approach to analyze high-frequency (up to 0.8 Hz) teleseismic records of deep earthquakes occurring in the Sea of Okhotsk. Our primary objective is to image the distinct features of P-wave velocity structure near the upper mantle discontinuity at the 660 km depth (660-D), at a spatial scale that remains unresolved in existing tomography models. Through rigorous modeling of global observations, we unveil the presence of an ultra-low-velocity pocket located beneath the subducted Pacific slab in the Sea of Okhotsk region. Building upon the background tomography results, we present a novel hypothesis: whether a deep-seated earthquake can evolve into a magnitude 8 or larger event hinges on the availability of sufficiently large stress, which serves as the trigger for thermal runaway processes that allow rupture to propagate outside of MOW. This stress likely stems from buoyancy induced by adjacent small-scale structure anomalies that have ultra-low velocity perturbation.

## Results

### Subevent analysis for the early rupture of the 2013 Mw 8.3 Sea of Okhotsk earthquake

To gain insights into the intricate structures associated with the 660-D depth in the vicinity of the subducting slab, we employ down-going P-waves from deep earthquakes to conduct a comprehensive subevent analysis. Our approach focuses on closely located earthquakes or sub-events within a significant earthquake, enabling us to leverage waveform data from one sub-event to calibrate and comprehend the waveforms from another, thereby unraveling intricate details that cannot be deciphered by analyzing a single event. However, the scarcity of intra-slab seismicity beyond a depth of 600 km poses significant challenges in terms of limited temporal and spatial data availability, hindering the use of multiple events to study fine scale features near the 660-D depth[17]. Additionally, variations in focal mechanisms between events add further complexity to the analysis. Consequently, larger events (e.g., Mw > 7.5) present more favorable opportunities for studying these phenomena. Firstly, the relative distance between sub-events within large deep earthquakes is smaller than the dimension of the earthquake itself, often measuring less than 100 km. Accurate determination of the relative source parameters (e.g., location and timing) of sub-events offers an opportunity to resolve structures at a scale comparable to the distances between sub-events. Secondly, the large magnitude of these earthquakes ensures a high Signal-to-Noise Ratio (SNR) for a vast majority of global broadband seismic array records, facilitating precise tracking of wavefield variations in distance and azimuthal profiles, thereby enabling a more robust discrimination between source and structural effects. Lastly, the dense sampling of ray paths near the seismic source permits discrimination between spatial variations in array waveforms caused by source-side structures or radiation patterns. This last point can be verified by comparing synthetic waveforms or array records from another nearby earthquake located a few hundred kilometers away.

Utilizing the aforementioned advantages and selection criteria, we have chosen the Mw 8.3 earthquake that transpired on 24 May 2013 in the Sea of Okhotsk region (Fig. 1a) as the target event for investigating its initial rupture process and the near source velocity structure beneath the Kurile Islands. This earthquake, documented in the global CMT (gCMT) catalog with a depth of 611 km[18,19], took place within the subducted Pacific slab in the northern Kurile subduction zone[5,20–23], it stands as the largest deep earthquake ever recorded and has been exceptionally well-captured by global arrays, providing us with a wealth of high-quality seismic data for our analysis.

We initiate our analysis by examining the sub-events associated with the Mw 8.3 earthquake. Waveform inversion and modeling of this event have indicated a rupture directivity towards 165°N along a sub-horizontal fault (e.g., refs. 5,20.). These findings have revealed the presence of four major sub-events (E1 to E4), as evidenced by the

Horizontal Directivity Parameter (HDP) displacement waveform record sections (Fig. 2a, c). While the direct waveform observations provide valuable insights, a clearer depiction of the sub-events is achieved with the Relative Source Time Function (RSTF) analysis (Fig. 2b, d). This discrepancy arises because displacement waveforms are dominated by long-period signals, while velocity waveforms encompass both positive and negative pulses. Given the inherent challenges in precisely modeling the observed waveforms at relatively high frequencies (0.1−0.5 Hz) on a wiggle-by-wiggle basis, this study focuses on the analysis and modeling of RSTF waveforms.

To derive RSFTs, we employ the Projected Landweber Deconvolution (PLD) technique[24,25] to deconvolve the 1-D synthetic waveforms from the observations (Supplementary Fig. 1). Synthetics are generated using the focal mechanism from the gCMT catalog and the Preliminary Reference Earth Model (PREM)[26]. Additionally, we utilize a path calibration technique to identify clean paths at the receiver side for RSTF modeling, employing a smaller nearby event (see Supplementary Note 2 and Supplementary Fig. 2 for details). The RSTFs are grouped and stacked within each rectangular grid of 3° (longitude) x 2° (latitude) to enhance the SNR. In our study, we focus exclusively on analyzing the initial sub-event and the early sub-event in the following rupture stage (E1 in Fig. 2d), as their waveforms are clear and exhibit minimal contamination. For ease of reference, we denote the initial sub-event as S1 and the early sub-event of E1 as S2 (indicated by dashed lines in the array records shown in Fig. 2 and Supplementary Fig. 3). Although subsequent sub-events E2-E4 are larger in magnitude, their source time functions are lengthier and more complex, with their waveforms being affected by coda waves from the preceding rupture. Consequently, utilizing them at relatively high frequency (0.1−0.5 Hz) poses significant challenges. Our findings demonstrate that even the initial portion of the data contains crucial information that contributes to an improved understanding of the earthquake rupture process and the velocity structure in the near-source region.

We then refine the relative timing and location of S1 and S2. The move-out of S2 in the USArray RSTFs is more discernible in the distance profile compared to the azimuthal profile, indicating a vertical offset between S2 and S1 (Fig. 3a, b). It is worth noting that a smaller sub-event (Sx) between S1 and S2 is also better observed in the distance profile (Fig. 3a, c). To generalize the distance profile analysis, we divide the global observations into different azimuthal groups (Fig. 3c), where positive move-out in the distance profiles is observed across all array profiles. This positive slope of S2 move-out indicates a larger ray parameter, suggesting that S2 is shallower than S1 with a vertical offset that is equal to or greater than the horizontal offset. The consistency in amplitude ratio between S1 and S2 across various azimuths strongly suggests similar focal mechanism of the two sub-events. This also diminishes the likelihood that S2 being a result of shallower reflection from S1 (see sensitivity test in Supplementary Fig. 14). In the global azimuthal profiles, both for station distances smaller and larger than 60°(Fig. 3e), sub-event Sx exhibits a more pronounced cosine move-out compared to S2, suggesting that the offset of Sx relative to S1 is primarily horizontal. However, the intermittent appearance of the Sx signal in azimuthal profiles implies that this sub-event may have a distinct focal mechanism compared to S1 or S2 (Fig. 3e).

The relative location and timing between subevents are determined by applying a grid search method on their arrival times (see more in Methods). The locations of S2 and Sx relative to S1 are presented in Fig. 3d and Table S1, revealing that Sx exhibits a sub-horizontal rupture directivity, while S2 ruptured sub-vertically. Sensitivity plots demonstrate that the relative location and origin time of these subevents are well-constrained (Supplementary Fig. 4). It is worth noting that the vertical offset between S1 and S2 of the earthquake was not reported in the published source models[5,20,27]. The subsequent larger subevents display clearer move-out in the azimuthal

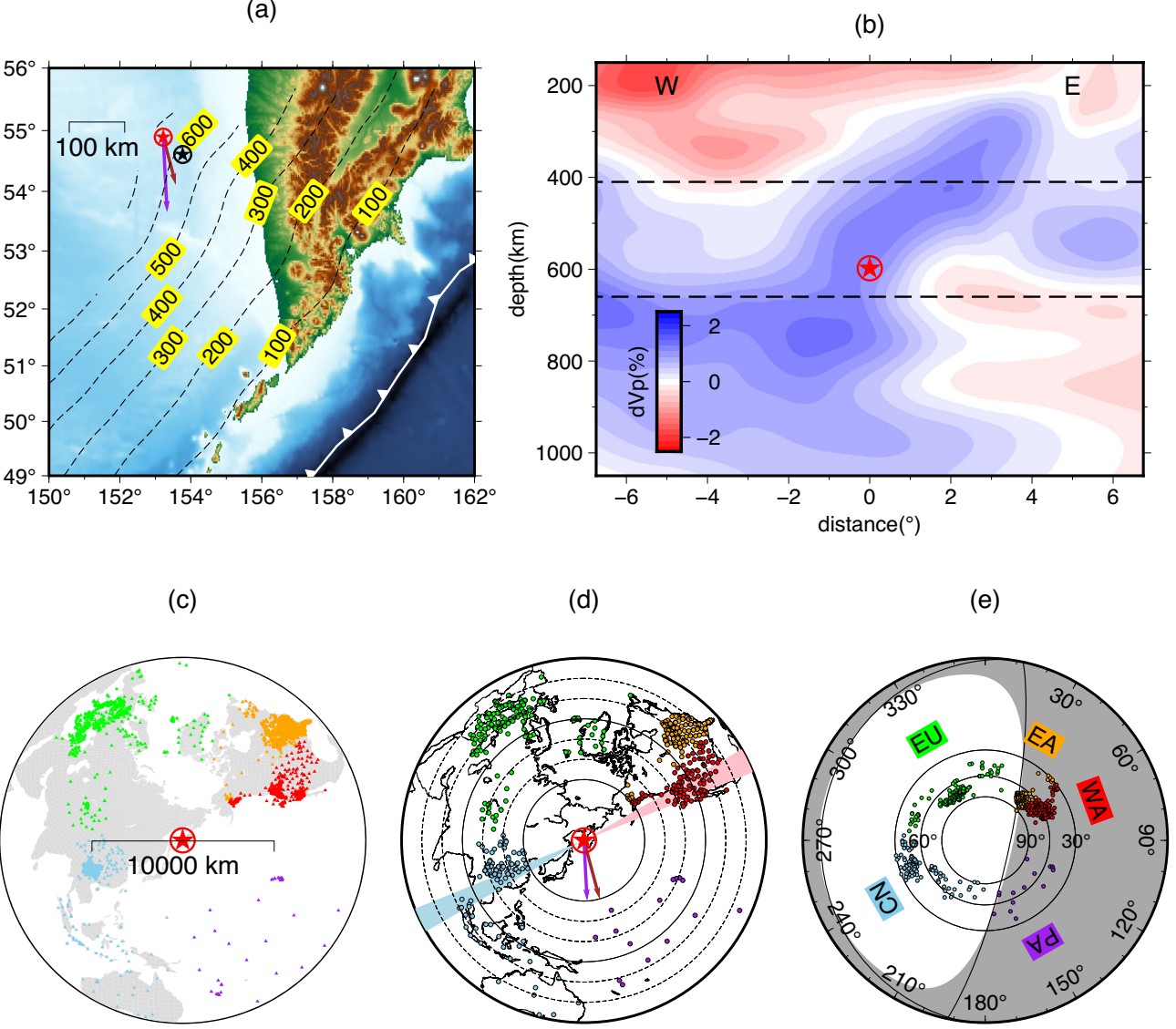

**Fig. 1 | Tectonic context, station distribution, and event characteristics.**
**a** Topographic map of the earthquake source region (Sea of Okhotsk). The dashed contours depict the Slab 1.0 model[56] with a depth interval of 100 km. The solid white line with triangles represents the location of the trench. The USGS hypocenter of the Mw 8.3 mainshock is denoted by a red star within a circle, while the centroid location from gCMT (globalCMT) is indicated by a black star within a circle. Brown and purple vectors illustrate the rupture extent along the strike directions indicated by refs. 5,19 and, respectively. **b** Two-dimensional P-wave tomographic profile from Fukao and Obayashi, 2013[34]. The upper and lower boundaries of the mantle transition zone (MTZ) are indicated by dashed lines. Letter "W" and "E" indicate the direction towards China and America, respectively.

**c** Global distribution of broadband stations (triangles) employed in this study. **d** Station coverage after waveform stacking. The cyan and pink shaded areas representing the azimuthal boundaries of the records utilized in Fig. 5. The brown and purple vectors maintain the same directions as in (**a**). **e** Moment tensor solution and the optimal double couple focal mechanism displayed on a beach ball. Stacked stations with corresponding colors are projected onto the beach ball. For panels (**c, e**), stations locations are categorized into five arrays based on their azimuths: CN (China/SE Asia array, azimuth: 180−275°, blue), WA (Western part of USArray, azimuth: 50−75°, red), EA (Eastern part of USArray, azimuth: 20−50°, orange), EU (European and central Asian stations, azimuth: 275−360° and 0−20°, green), and PA (Pacific island stations, azimuth: 90−180°, purple).

profile than in the distance profile (Figs. 2 and S3), likely due to their larger horizontal offsets in comparison to any vertical offsets, if present[21].

## In-plane multipathing of S2 in North America
At the frequency range of 0.1−0.5 Hz, subevent S2 in the HDP RSTF record sections exhibits consistent waveforms across the North American continent (Fig. 2d). However, distinct features emerge when dividing the USArray into the EA array (azimuth range: 25 to 50°) and the WA array (azimuth range: 50 to 75°). Despite that the EA array sample the beachball closer to the nodal plane (Fig. 1e), the S2 waveforms remain relatively consistent across the distance profile (Fig. 3c). On the other hand, the S2 waveforms in the WA array display greater

complexity (Fig. 3c). To gain a better understanding of these waveforms, we further divide the WA and EA arrays into 2 and 4 azimuthal bins, respectively (Fig. 4c).

For the 2 azimuthal bins in the EA array, the S2 waveforms consistently align in the respective distance profiles. In contrast, the WA array exhibits considerable variations in the WA3 and WA4 profiles, where the peak amplitude arrivals of S2 exhibit a "jump" around distances of 55° to 60°, with an offset of approximately 1 s. This phenomenon is known as the "multipathing effect," which is typically induced by abrupt velocity changes near the core-mantle boundary along the ray path[28,29]. Recent studies have further unveiled the presence of such sharp velocity changes within the mantle transition zones[30−32]. As this feature is clearly observed in the distance profiles

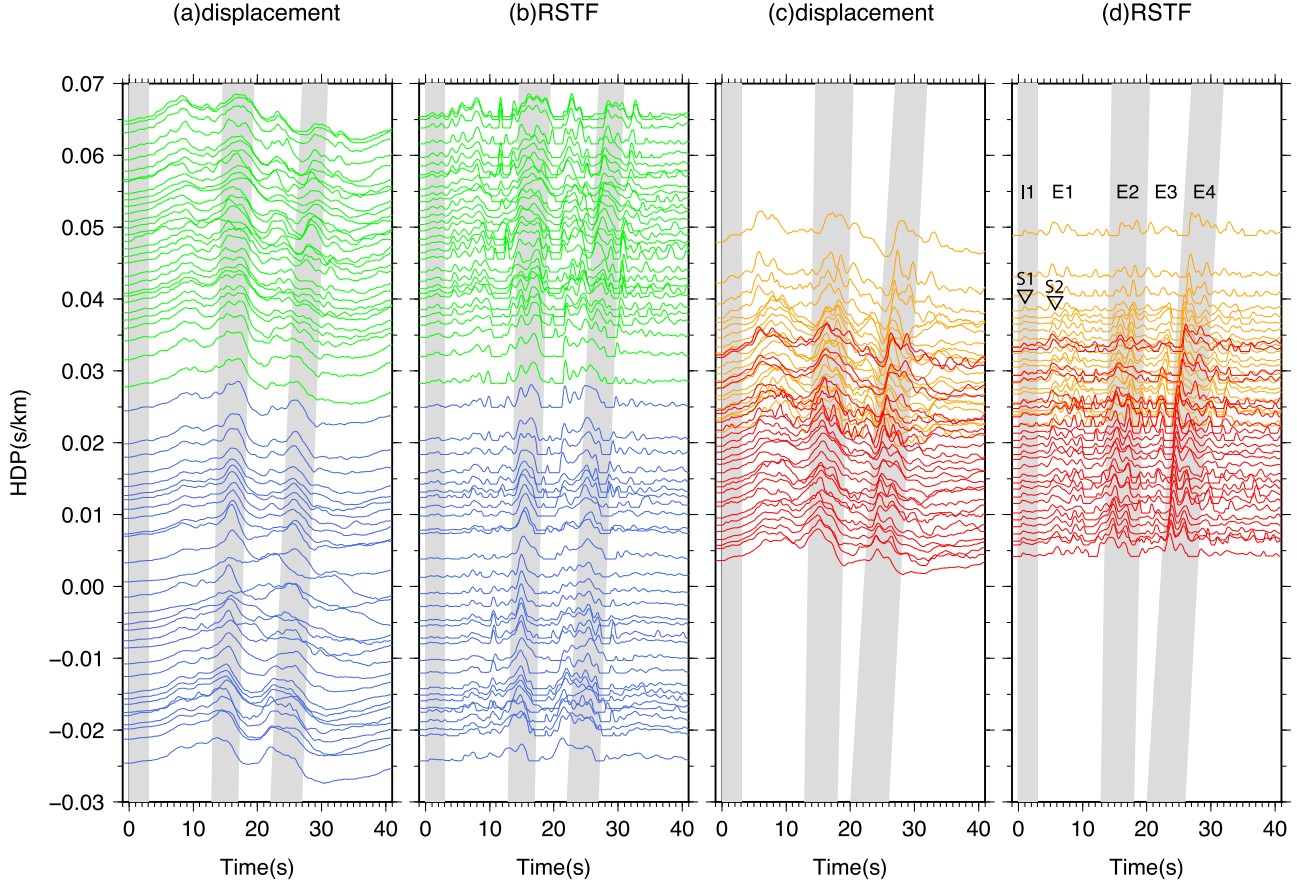

**Fig. 2 | Stacked displacement waveforms and corresponding deconvolved RSTFs (relative source time functions). a** Stacked displacement waveforms of EU (European and central Asian stations, green) and CN (China/SE Asia array, blue) array plotted against HDP (Horizontal Directivity Parameter). The gray belts highlight subevents I1, E2, and E4 (see (**d**)). **b** RSTFs for (**a**) derived at 0.1–0.5 Hz. **c, d** Similar to (**a, b**), but for the data obtained from USArray, with the orange color representing EA (Eastern part of USArray) and the red color representing WA (Western part of USArray). The first pulses of subevents I1 and E1 are marked as S1 and S2, respectively.

rather than in the azimuthal profiles, we therefore attribute it to in-plane multipathing[28,30,33]. Furthermore, the positive slope of the moveout in the stance profile gradually diminishes at greater distances (e.g., beyond ~75° as in WA2 profile in Fig. 4b), indicating that the effect of vertical directivity has been counteracted by the velocity anomaly.

To further investigate the in-plane multipathing phenomenon of S2, we employ the "multipathing detector" (MPD) to analyze the S2 RSTF waveforms[30]. The MPD measures the splitting time and amplitude ratio between the two pulses generated by multipathing. The results of the multipathing detection on the 0.2−0.8 Hz RSTF waveforms are presented in Fig. 4c as map views. Generally, the amplitude ratio between the first and the second pulses decreases as the epicenter distance increases.

Notably, these coherent multipathing features are primarily observed in the S2 waveforms, indicating that the location of the velocity anomaly is likely near the seismic source rather than in the intermediate region or beneath the stations. Importantly, it should be emphasized that the S1 waveforms exhibit much greater stability and do not display multipathing characteristics in the distance profiles (Fig. 4a, b). This contrasting behavior suggests that the origin of the S2 multipathing phenomenon stems from a small-scale velocity structure anomaly in close proximity to the seismic source, which mainly affects the ray paths of S2.

**Subslab ultra-low-velocity anomaly induced multipathing**
The ray paths towards North America mainly sample the structure beneath the subducting slab, where a subtle low-velocity anomaly is present beneath the Pacific slab in the tomography model (Fig. 1b). The velocity structure associated with this low-velocity anomaly potentially generates the multipathing features observed for S2. However, in the existing tomography model[34], the amplitude of the P-wave velocity anomaly in this low-velocity structure is only − (2−3)%, which is insufficient to produce the observed multipathing effects. To further investigate the multipathing effect observed for S2, we construct 2-D velocity models comprising three hypothetical seismic features in the source region (Supplementary Fig. 5), referred to as "SUP" models, where "S" represents slab anomaly, "P" represents background plume anomaly and "U" represents Subslab Ultra Low-Velocity Anomaly (SULVA). Synthetic waveforms generated from different combinations of "SUP" features are used to demonstrate their sensitivities to waveform complexity (Supplementary Fig. 5). Among various combinations, only the models with "U" structure can reproduce the multipathing feature of S2 (Supplementary Fig. 5b), while the effects of the slab and plume can be disregarded. Therefore, we propose that a small-scale SULVA is responsible for generating the multipathing of S2. Such small-scale velocity anomalies often elude detection in tomography images due to the smoothing and coarse grid size in tomography inversions.

Next, we determine the location, geometry, and amplitude of the velocity reduction associated with the SULVA through forward waveform modeling. Our goal is to replicate the observed multipathing features in the RSTF waveforms. Since the multipathing of S2 primarily occurs in-plane, we expect that synthetic seismograms generated from 2-D velocity models will capture the main characteristics of the

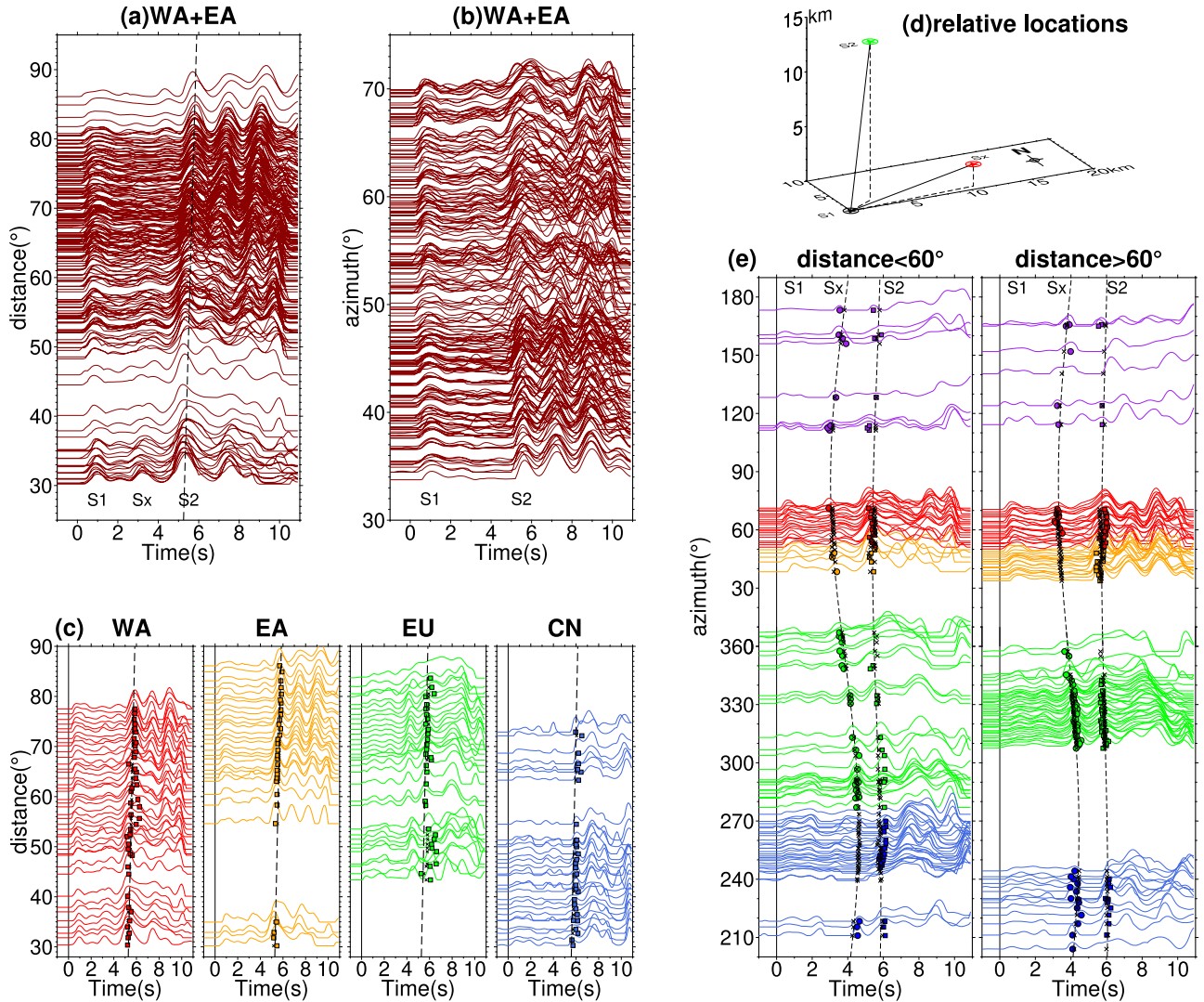

**Fig. 3 | Relative timing and location analysis of sub-events based on RSTFs (relative source time functions). a** Distance profile of RSTFs derived at 0.1–0.5 Hz for the WA (Western part of USArray) and EA EA (Eastern part of USArray) arrays, showing three sub-events: S1, Sx, and S2. **b** Azimuthal profile of RSTFs showing the same sub-events as in (**a**). **c** Distance profile with downsampled RSTFs for the EU (European and central Asian stations), CN (China/SE Asia array), WA, and EA arrays, with dashed lines representing the arrivals of S2. **d** Optimal locations of Sx (red star) and S2 (green star) relative to S1 (black star). **e** Azimuthal profiles with a distance range split at 60°. Black dashed lines indicate the predicted arrival time for sub-events Sx and S2, assuming distances at 45° and 75°, respectively. Circles and squares represent the picked arrivals for Sx and S2, respectively, while crosses indicate the predicted arrivals from the optimal rupture model.

wavefield. To achieve this, we utilize the same PLD method employed for the real data to deconvolve the synthetic seismograms using the same 1-D Green's functions. By comparing the synthetic RSTF with the observed data, we identify the best-fitting SULVA model. The 2-D synthetic seismograms are generated using a GPU version of a 2-D finite-difference code[35], which is much more efficient than the 3D simulations. We select stations within a pair of narrow azimuthal ranges, specifically WA3 and its corresponding opposite azimuthal bin in the CN array (Fig. 1d). This approach allows us to employ a single 2-D velocity model to simulate the observed RSTFs for both directions simultaneously (see more in Methods).

The primary cause of the multipathing feature observed in S2 is the splitting of the seismic wave energy along the boundary of the SULVA. The second branch of the multipathing phase arrives later at larger distances (e.g., Fig. 5c), indicating that the ray path of S2 samples the upper boundary of the SULVA. On the other hand, the ray path of S1 either does not sample or entirely penetrates through the central region of the SULVA, resulting in no multipathing for S1. Given the proximity of S1 and S2 (less than 20 km apart), the scenario where S1

penetrates through the middle part of the SULVA is more likely (Supplementary Figs. 5a, 6).

To parameterize the SULVA structure, we introduce a model called the "cut-donut" (Fig. 5a). This model has a ring-shaped structure with inner and outer radius of r and R, respectively. This design aims to place the ray path of S1 penetrating through the central part of the SULVA. We assume a uniform reduction in P-wave velocity within the SULVA, as an ultra-low-velocity structure with a scale of a few tens of km at this depth are more likely to be produced by chemical or compositional anomalies rather than gradual thermal changes[30,33,36].

To evaluate the quality of the fit between the model prediction and observed RSTFs, we assess the relative timing between S1 and S2, which is sensitive to the origin time difference and relative location between them (Fig. 5b, c). Additionally, for the WA3 profile, we compare the waveform similarity between the data and synthetics. To simultaneously fit the waveform shape and arrival time, we utilize a "template matching" cross-correlation (TMCC) as an indicator of the consistency between the data and synthetics (Methods). The TMCC

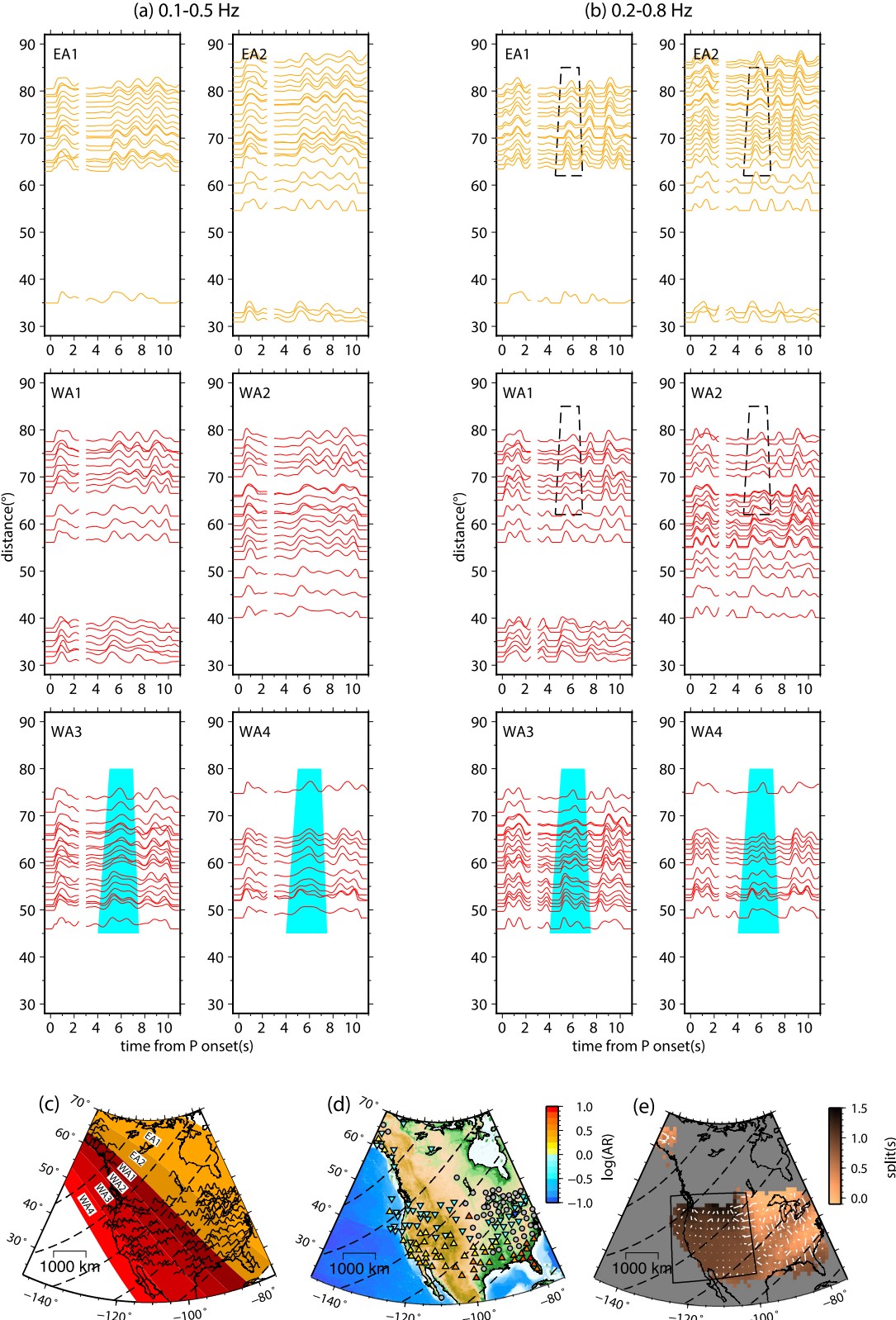

analysis allows us to quantify the similarity between the observed RSTFs and the synthetic waveforms. The waveform cross-correlation coefficients are used to determine the degree of consistency between the model predictions and the observed data. This approach enables us to optimize the parameters of the "cut-donut" model, such as the radius of the inner and outer circles, to achieve the best fit to the observed RSTFs.

Among the five parameters, the modeling is the most sensitive to the dipping angle of cut2 and the velocity reduction. A change of more than 5° in the dipping angle or a bias of 4% in the optimal Vp reduction can dramatically alter the multipathing pattern (Fig. 5b and Supplementary Fig. 7). On the other hand, variations in cut1 dipping angle do not introduce a strong impact, as expected, since the left boundary is away from the distance range where the multipathing occurs. The

**Fig. 4 | RSTF (relative source time functions) record sections further divided by smaller azimuthal bins: EA1: 24-42°, EA2: 42-50°, WA1: 50-55°, WA2: 55-60°, WA3: 60-68°, WA4: 68-74°. a** Deconvolution results obtained from bandpass filtering between 0.1 and 0.5 Hz. **b** Deconvolution results obtained from bandpass filtering between 0.2 and 0.8 Hz. Subevents S1 and S2 are depicted separately with consistent amplification normalization. Cyan shading in distance profiles WA2 and WA3 indicates in-plane multipathing observations, with additional multipathing effects highlighted within the dashed box. The data is down-sampled to a maximum of one trace per 1° distance range. **c** Azimuthal bins of sub-arrays indicated by shaded colors, with RSTFs of 0.2–0.8 Hz plotted on the map. Dash line contours represent 10° intervals of epicentral distances. **d** Map view displaying the logarithms of amplitude ratio for the second pulse relative to the first pulse of the split at 0.2−0.8 Hz. Gray circles represent stations with a single pulse (no multipathing). **e** Observation of multipathing arrival splits for subevent S2 observed in the USArray. The region with significant splitting is enclosed by solid black lines.

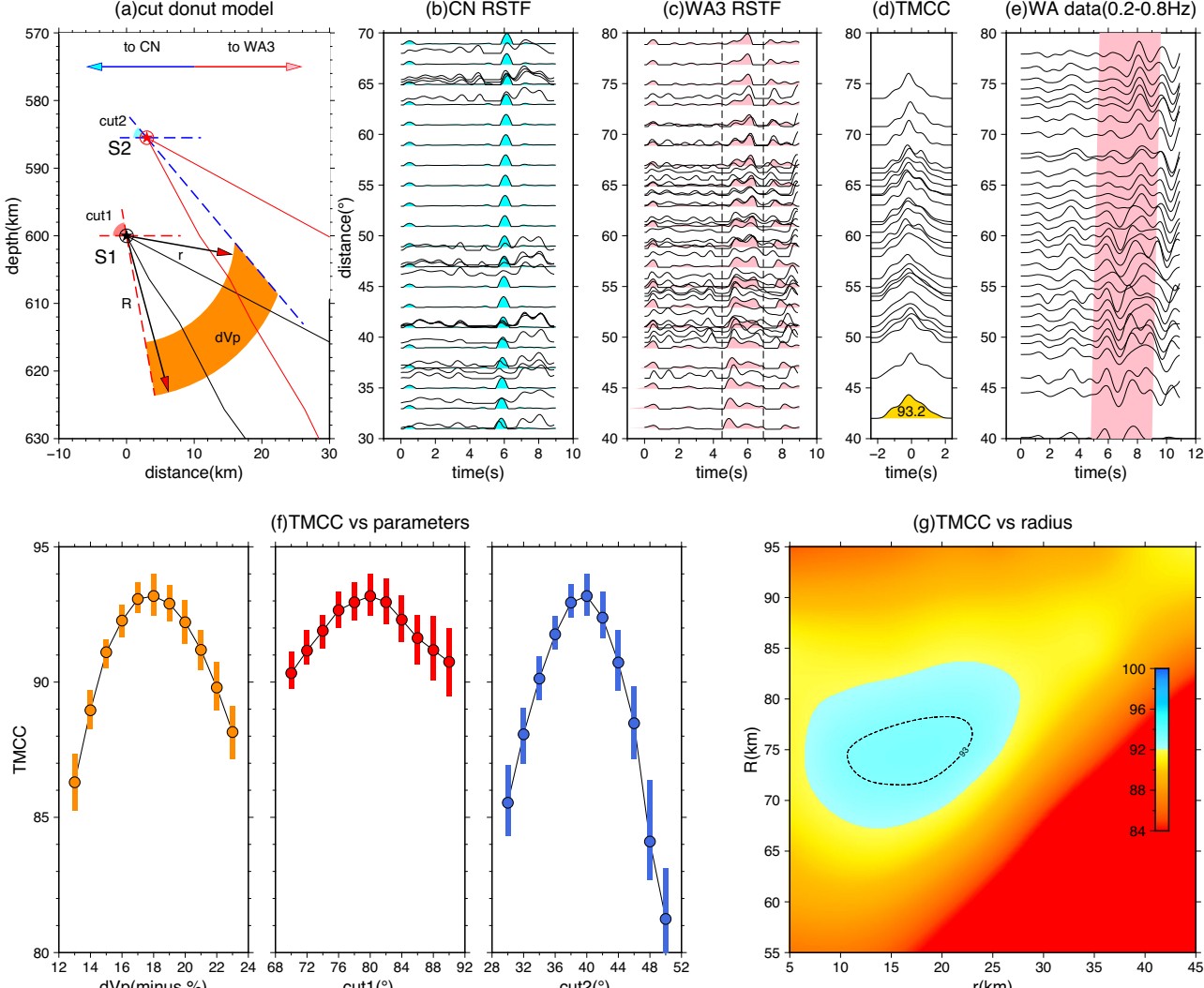

**Fig. 5 | High-frequency waveform modeling of "Cut donut" shaped SULVA (subslab ultra-low-velocity anomaly). a** Depth profile of the structure model illustrating the relative location between the SULVA and subevents S1/S2. The SULVA structure is defined by five parameters: r and R for the inner and outer curve radius from S1, respectively; cut1 for the dip angle of the left boundary from S1; cut2 for the dip of the right sharp boundary from S2, shown as dash lines in a "scissor" shape. The teleseismic ray path range predicted by the PREM (Preliminary reference Earth model) model is indicated by black and red solid curves. **b** Synthetic RSTFs (relative source time functions) obtained from modeling with the optimal SULVA, overlapped with observed RSTFs on the CN (China/SE Asia array) array. **c** Same as (**b**), but for the WA3 subarray. The dash lines indicate the time window used for calculating TMCC (template matching cross-correlation). **d** TMCC results between each observed RSTF and the corresponding synthetics after Akima interpolation, with the maximum cross-correlation (CC) value listed on the left. The shaded area in gold represents the cross-correlation summation along the time window, and the maximum normalized CC value is marked below. **e** Velocity data of the WA array filtered between 0.2 and 0.8 Hz before stacking, highlighting the strong multipathing effect in pink. **f** Error estimation of three parameters (velocity perturbation dVp, dip of the left boundary cut1, and right boundary cut2) for the cut-donut model. The vertical bars indicate the range for 95% of TMCC values after 2000 iterations of bootstrapping-style random station picking. **g** Two-dimensional plot of TMCC values with corresponding r and R values, showing the parameter space exploration.

errors of dVp and the dipping angles of cuts are estimated using a bootstrapping method, which involves randomly sampling 80% of the data 2000 times to define a 95% confidence interval. For the WA3 subarray, we determine that dVp = 18 ± 2% and cut2 = 40 ± 2°.

To determine the values of r and R, we conduct a 2-D grid search. Although there are some trade-offs, the grid search provides a good constraint on R (~75 km). The inner circle radius r is less well constrained, with the optimal values ranging from 10 to 25 km (Fig. 5g and

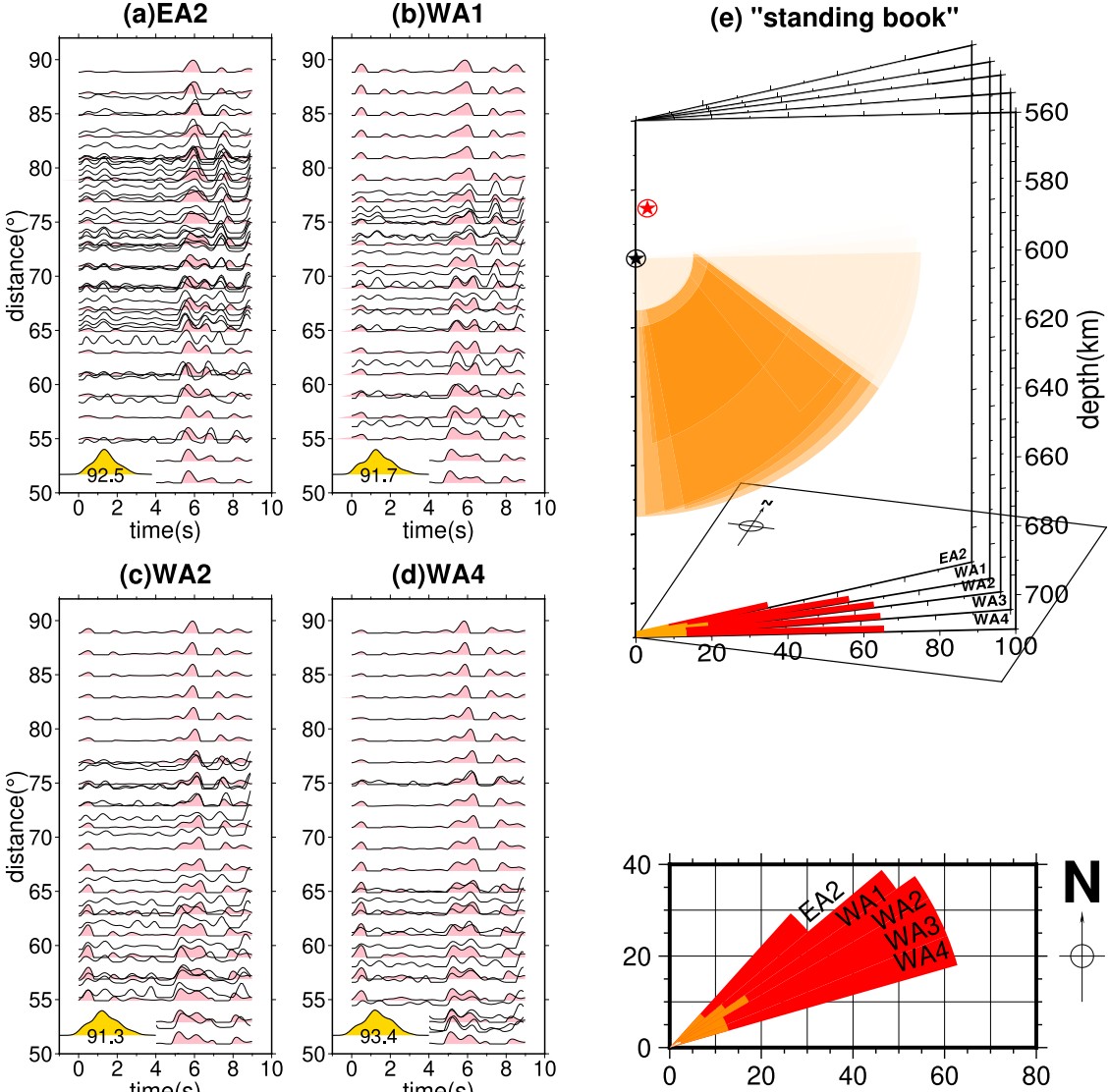

**Fig. 6 | Modeling of Optimal SULVA (subslab ultra-low-velocity anomaly) Structures for Different Azimuthal Bins. a–d** Comparison of RSTF (relative source time functions) waveform fits between observed data (black traces) and synthetics (pink) generated from the optimal models listed in Table S2. Corresponding TMCC (template matching cross-correlation) traces are plotted at the lower left. **e** Visualization of the "Standing Book" representation for the cut-donut model of the SULVA structure for the five sub-arrays. The orange-colored parts indicate the extent of the SULVA structure, while the color-faded regions represent the discarded "cut" parts based on parameters r and cut2. The black and red stars indicate the location of subevents S1 and S2, respectively. The inner and outer curves of the SULVA are projected onto the horizontal surface below, colored in dark orange and red, respectively.

Supplementary Fig. 8). We apply the same approach to other North American distance profiles, except for EA1 due to its small range of distance coverage. The optimal models for these five profiles are listed in Table S2. Figure 6 shows the RSTF waveform fits and the relative location of the SULVA at EA2, WA1, WA2, and WA4. With data in an azimuthal range of 42−74°, we determine that the SULVA has a horizontal dimension of 30 × 60 km and a depth range of approximately 60 km (Fig. 6e). Both its horizontal and vertical dimensions gradually reduce by approximately 50% from southeast to northwest. The velocity perturbation remains stable in all the profiles, with a P-wave velocity reduction of 18−20%.

## Discussion

The robustness of the SULVA was evaluated by synthetic tests on other parameters and scenarios, including anisotropy, attenuation, the ratio of velocity perturbation between P and S waves, out-of-plane multipathing effects, and variations in the focal mechanism of subevents.

The results of these tests indicate that these parameters and scenarios have limited impacts on the multipathing effects, except in some extreme cases (Supplementary Note 2, Supplementary Figs. 9−13). Additionally, we conducted tests to evaluate the potential influence of the 660-D topography. Our tests revealed that while the 660-D topography has the capability to induce uniform changes in arrival, it falls short in replicating the observed multipathing pattern (Supplementary Fig. 16). Hence, we conclude that the source side SULVA obtained from our analysis is robust.

At the sites of three modern M8+ deep earthquakes, 2-D velocity profiles from a representative global P-wave tomography model[34] all show the existence of low velocity anomaly beneath the subducted slab (Fig. 7a, d, f). Independently, a higher-resolution regional travel-time tomography image revealed a clear low velocity structure with even larger P-wave velocity reduction directly beneath the source of the 2018 M8.2 Fiji earthquake[37] (Fig. 7h). Also note that this low velocity structure has distorted the geometry of the subducted slab in a way

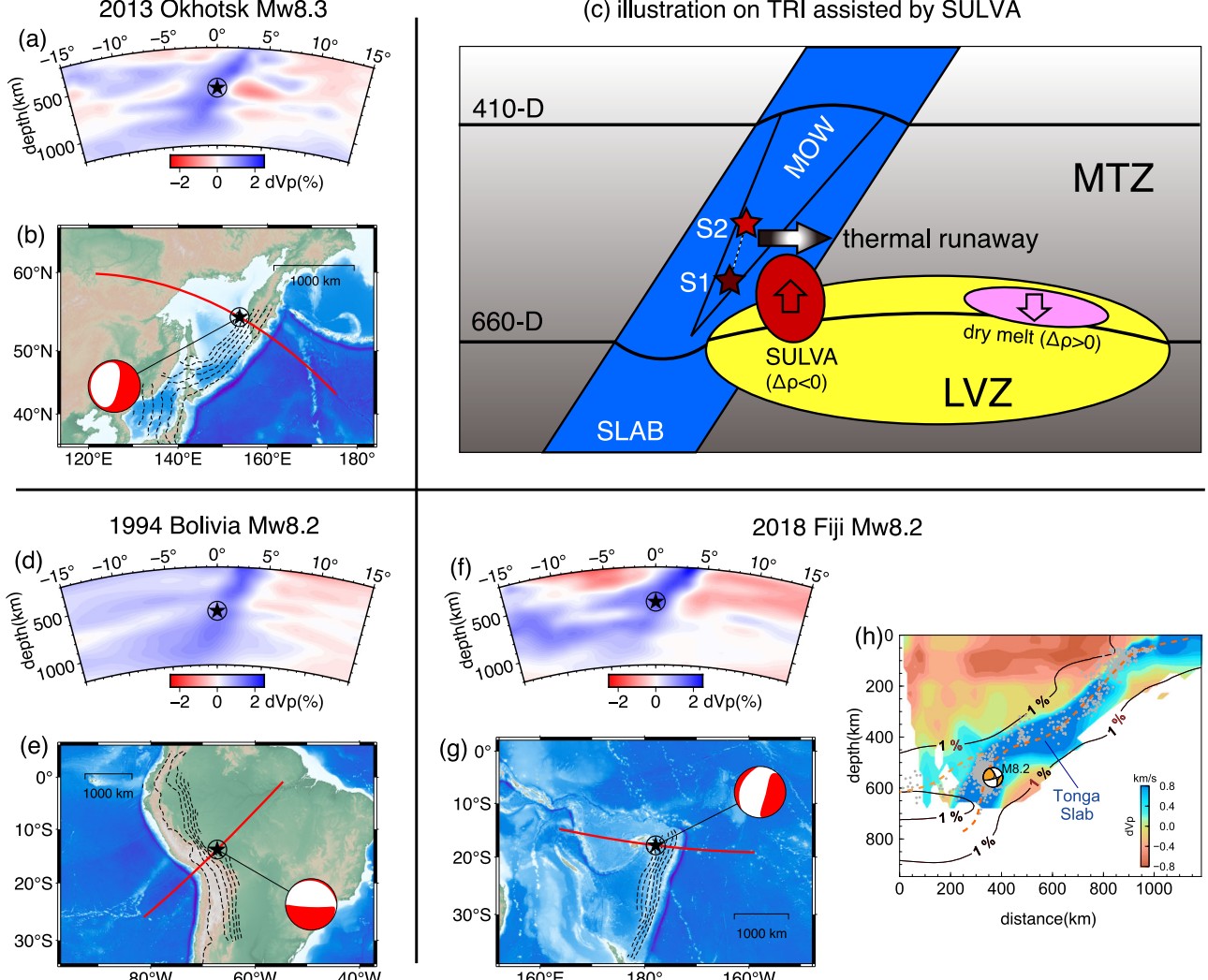

**Fig. 7 | Schematic illustrations supporting the hypothesis of the SULVA (sub-slab ultra-low-velocity anomaly) triggering large deep earthquakes.**
**a** Tomography images[34] depicting the 2013 Okhotsk earthquake. **b** The map view of the source region with gCMT (globalCMT) focal mechanisms. The star marks the gCMT centroid's location, and the red line represents the surface projection of the tomographic profile. The dashed contours depict the Slab 1.0 model[56] with a depth interval of 100 km. **c** A conceptual demonstration illustrating the mechanism triggering TRI (thermal runaway instability), leading to ruptures extending beyond the cold slab core (MOW, metastable olivine wedge). Within the volatile-bearing SULVA structure (red ellipse), substantial buoyancy remains positive across the 660-D, as

indicated by the upward arrow. The initial rupture process (S1-S2) takes place within the MOW, but it evolves into a TRI, aided by the buoyancy stress originating from the SULVA. While the density of dry melt (pink ellipse) is higher than the ambient mantle after crossing the abrupt crystalline phase change boundary. **d, e** Similar to (**a**, **b**) but for the 1994 Bolivia earthquake. **f, g** Similar to (**a**, **b**) but for the 2018 Fiji earthquake. **h** Modified from Jia et al.[57]. The 1% P-wave velocity anomaly contours derived from the GAP_P4 model Fukao and Obayashi[34], are represented by solid black lines. The background colors illustrate the regional tomography model developed by Conder and Wiens[37]. The gray dots denote background seismicity.

that the velocity anomaly exerts a buoyance beneath the slab. Such consistency suggests that the interaction between high-temperature structures, shown as low velocity anomaly in tomography models, and the subducting slab within the MTZ could give rise to the SULVA, which in turn facilitates the rupture of M8+ deep events. The sub-slab low-velocity anomalies observed in tomography models can originate from either a mantle plume[38] or the entrainment of upper mantle material during subduction[39,40]. The presence of these low-speed velocity structures suggests a local high temperature anomaly, which is conducive to the incubation of the SULVA.

The remarked reduction in SULVA velocity may arise from partial melting processes associated with volatile compounds[41,42]. The MTZ is recognized for its significant hydration[43–45], particularly in conjunction with water influx from subduction processes, rendering the lower portion of MTZ as potentially the most 'moist' region within the Earth's mantle[43,46]. The inclusion of volatile constituents, such as water, can

notably lower the solidus temperature compared to a system devoid of volatiles[47,48]. Following melt generation, gravitational forces lead to the separation of the melt from the remaining solid materials[47,49]. However, the density contrast between these volatile-bearing melts and the residual solids under MTZ conditions is intricate[46,50]. The density of the volatile-free melt is lower than that of the surrounding mantle below the 660-D phase-transition boundary. Yet, as the melt traverses this boundary, it undergoes a density increase, potentially resulting in entrapment or stagnation at the base of the MTZ[46,51] (Fig. 7c, pink ellipse). Conversely, if the melt contains a substantial concentration of water, its density can consistently remain lower than that of the surrounding solid material, both above and below the 660-D[46] (Fig. 7c, red ellipse).

The SULVA we imaged shows its presence both above and below the 660-D (Fig. 6e). First-principle calculations and lab experiments suggest that the density contrast between melt and the ambient solid

mantle is generally larger below 660-D compared to that above 660-D. This is attributed to the abrupt mineral phase change that occurs at the 660-D for solid materials[50,52–54], as the mineral phase change across the 660-D is abrupt[46,53]. For a $(Mg_{0.75},Fe_{0.25})_2SiO_4$ melt with 4 wt% water, the density is approximately 3920 kg/m³. Above and below the 660-D, the densities of the solid mantle are approximately 4000 and 4400 kg/m³, respectively[46,50]. Consequently, as the melt crosses the 660-D, there is a significant change in density contrast between the solid mantle and the melt, ranging approximately from 80 to 480 kg/m³. Based on the derived optimal SULVA structure, assuming a vertical extent of approximately 60 km, we can estimate the maximum stress induced by the buoyancy of the SULVA. If the depth of 660-D has not been elevated, then ~10 km of the lower portion of the SULVA is immersed in the lower mantle, resulting in a buoyancy stress of approximately 86 MPa. However, if the 660-D has been elevated by 10 km due to higher temperatures, the buoyancy stress would increase to approximately 125 MPa. Similarly, when the SULVA structure consists exclusively of volatile materials, such as diamonds, one should take into account its density at the base of the MTZ, which is approximately 3700 kg/m³[55]. In the case of a flat 660-D, the corresponding buoyancy stress measures approximately 216 MPa. However, if 660-D has been uplifted by 10 km, the buoyancy stress increases to around 255 MPa (see more details in Methods).

This significant buoyancy and its proximity to the epicenters of large deep earthquakes provides valuable insights into the rupture mechanism of these events. The distortion of the slab near the 2018 Mw8.2 Fuji deep event further supports such buoyancy effect (Fig. 7h). The conventional transformational faulting within the MOW region alone is insufficient to account for the observed large rupture widths in M8+ earthquakes[5,20]. Furthermore, the preferred rupture direction of the Mw8.3 Sea of Okhotsk earthquake does not align well with the slab model based solely on seismicity data[32,56]. Thus, the "dual-mechanism" model is proposed that the occurrence of the most significant deep-seated earthquakes consists of two consecutive stages: the initial rupture commences within the cold, MOW located in the core of the subducting slab, and subsequently propagates to the outer regions[9,57]. The most widely accepted theory for explaining the latter phase is the TRI hypothesis[3,6,9]. However, numerical modeling simulating conditions at intermediate depths and pressures suggest that the critical shear stress required to trigger TRI is on the order of 0.8−2 gigapascals[7], along with earthquake stress drops of several hundred megapascals[58]. These values are significantly higher than those derived from seismological observations[9,58]. Alternatively, considering rheological mechanisms such as diffusion creep, dislocation accommodated grain boundary sliding, and low-temperature plasticity can lower the required peak stress to a range of 100−300 MPa[8]. As described in the preceding paragraph, the SULVA structure, potentially associated with partial melting and/or volatiles, could provide the additional stress required to support thermal instability, allowing the initial rupture within the narrow MOW to propagate out of MOW and develop into M8+ earthquakes. A critical factor in this mechanism is the density contrast between the SULVA and the surrounding mantle, which generates buoyancy-induced stresses[59]. These findings indicate that the interaction between water-rich melts and the subducting slab, along with its position relative to the 660-D, may serve as a prerequisite for the formation of the TRI. Actually, all M8+ deep earthquakes occurred near the 660-D suggest that this mechanism is likely universal in facilitating the formation of such events. Therefore, the occurrence of a large deep earthquake is primarily determined by the thermal and velocity structure of the surrounding mantle, rather than the intrinsic physical properties of the slabs themselves. Further interpretation of such enormous velocity reduction near 660-D may require further mineral physics and geodynamic investigations.

## Methods

### Waveform data processing

We retrieved global teleseismic P-wave data within a distance range of 30 to 90°. Subsequently, this dataset was segmented into distinct azimuthal bins. Quality control and initial phase arrival picks were manually conducted, followed by a more precise arrival time picking by cross-correlation techniques. We then downsampled data to 0.1 s sampling interval to allow easier processing of the dataset. To derive RSTF for the selected data, we computed the 1-D synthetics using the PREM and GCMT solution of the mainshock (with 1 s source time function) then deconvolved them from the observed waveform data. Here the PLD deconvolution method[34] is used to ensure that there is no negative phase in the RSFT. The time window for deconvolution is selected to be 2 s before and 15 s after the P-wave arrival. Since some of the arrays have very dense station distribution, to enhance signal to noise ratio and to reduce the number of waveforms for better visualization, we stacked the RSTFs with a rectangle area defined by 2 degrees in latitude and 3 degrees in longitude. The RSTFs were then filtered to various frequency bands for further studies, e.g., key feature identification and modeling. Both the raw waveform data and final RSTFs could be found in the "Data availability" section.

### Relative relocation between sub-events

At a specific station, the arrival time difference (T) between a subevent (e.g., Sx, S2) and the reference subevent (e.g., S1) is represented by the following equation:

$$T = \tau + \Gamma * L + \Delta * H \qquad (1)$$

Here, $\tau$ is the origin time difference between subevents. The parameter $\Gamma$ denotes the horizontal rupture directivity parameter, which is calculated as $\Gamma = -\cos(\Theta)*\sin(\Phi)/v$. Here, $\Theta$ represents the azimuth of the station relative to the rupture direction, $\Phi$ signifies the take-off angle, and $v$ corresponds to the P-wave velocity. Additionally, $\Delta$ is the vertical rupture directivity parameter, given by $\Delta = -\cos(\Phi)/v$. The parameters L and H represent the horizontal and vertical offsets, respectively.

To identify the optimal values for the origin time difference, horizontal offset, and vertical offset, we employ a grid search method. This method systematically explores various combinations and select those that minimize the following misfit function:

$$err = \sum_{i=1}^{N} (\tau + \Gamma_i * L + \Delta_i * H - T_i)^2 \qquad (2)$$

In our analysis, N represents the total number of stations. The variables $T_i, \triangle_i, \Gamma_i$ refer to the observed delay time, vertical rupture directivity parameter, and horizontal rupture directivity parameter (as defined in Eq. (1)), respectively, corresponding to the i-th station. To minimize the error (as shown in Supplementary Fig. 4), we conduct a grid search over the parameters $\tau$, $L$, $H$, and $\Theta$.

### 2-D FD waveform modeling

Given that the predominant feature in our observations is in-plane multipathing rather than out-of-plane multipathing, 2-D simulations are sufficient to capture the key features. We employed a 2-D finite difference (FD) code[33] to simulate synthetic waveforms, which were then deconvolved from the same 1-D synthetics as we applied to data to derive the synthetic RSTFs. The best 2-D velocity model was determined by comparing the observed and synthetic RSTFs. The input 2-D velocity model could either be a tomography model, or be manually designed based on needs. Earth-flattening was applied to account for the curvature of the Earth in our 2-D model implementation. Given that the highest frequency applied in our study is 0.8 Hz and considering the lowest P velocity in the model is 5.8 km/s, the

corresponding minimum wavelength is calculated to be 7.25 km. To adhere to the Nyquist theorem and alleviate spatial dispersion resulting from discretization, it is advisable to set the grid dimension to be smaller than 1/6 of the minimum wavelength[33]. Consequently, we have opted for a grid size of 1 km to appropriately balance these considerations. Due to the nature of staggered-grid in the FD code, it is quite straightforward to implement a new 2-D model in the simulation, which is critical for our study, as a large amount of 2-D models were tested. The FD code is written by CUDA and optimized for GPUs, which dramatically shortens the computational time compared with traditional CPU implementation. For instance, in our test, with three V100 SXM2 GPU cards and simulations to teleseismic distances, it only takes 9 min to generate the synthetics with numerical accuracy of 1.5 Hz. This high efficiency is also important for us to test a large numbers of models. Sensitivity tests from 2-D simulations are very helpful to understand the wavefield propagation and construct a reliable 3-D model when it is needed. In general, the design of the 2-D models is based on the understanding and sometimes speculation on the features in the observations, e.g., the multipathing effects as we identified in the RSFTs.

### "Template matching" cross-correlation (TMCC)

A time window of 2.5 s for the S2 RSTF is selected, and it is cross-correlated (CC) with the data using a moving time window approach. At each time mark in the profile, the cross-correlation value is calculated and summed across all pairs of traces. The Total Moving Cross-Correlation (TMCC) value for a specific model is determined by identifying the peak value of the summation of all cross-correlation traces after normalization. This process is illustrated in Fig. 5d.

### SULVA induced buoyancy when spanning across 660-D

The stress resulting from buoyancy can be expressed in a simplified form as follows:

$$\sigma = \Sigma \Delta\rho_i g h_i = \Delta\rho_U g h_U + \Delta\rho_D g h_D \qquad (3)$$

Here $\Delta\rho_U$ and $\Delta\rho_D$ represent the density contrast between SULVA and the ambient mantle above and below the 660-D boundary, respectively. Similarly, $h_U$ and $h_D$ denote the depth extent of SULVA within the upper and lower mantle, respectively. Hence, in the scenario of $(Mg_{0.75}, Fe_{0.25})_2SiO_4$ melt with 4 wt% water, where the 660-D has not been uplifted, the stress induced by buoyancy can be expressed as follows:

$$\sigma = (80\,kg \cdot m^{-3} \times 50\,km + 480\,kg \cdot m^{-3} \times 10\,km) \times 9.8\,m \cdot s^{-2} = 86\,MPa \quad (4)$$

If the 660-D has been uplifted by 10 km because of the local thermal anomaly, the subslab stress induced by buoyancy can be calculated as follows:

$$\sigma = (80\,kg \cdot m^{-3} \times 40\,km + 480\,kg \cdot m^{-3} \times 20\,km) \times 9.8\,m \cdot s^{-2} = 125\,MPa \quad (5)$$

**If SULVA is purely consist of volatiles, for example, diamonds, if 660-D is not been lifted, then the buoyancy stress is:**

$$\sigma = (300\,kg \cdot m^{-3} \times 50\,km + 700\,kg \cdot m^{-3} \times 10\,km) \times 9.8\,m \cdot s^{-2} = 216\,MPa \quad (6)$$

**If 660-D has been lifted by 10 km, then the buoyancy stress is:**

$$\sigma = (300\,kg \cdot m^{-3} \times 40\,km + 700\,kg \cdot m^{-3} \times 20\,km) \times 9.8\,m \cdot s^{-2} = 255\,MPa \quad (7)$$

## Data availability

The synthetic Green's function data generated in this study and raw waveform have been deposited in the Mendeley database under accession code https://data.mendeley.com/datasets/3hvttt7rgy/. The waveform data can be accessed by the IRIS DMC. The CNSN waveform data can be retrieved by the SEISDMC at Institute of Geophysics, China Earthquake Administration (https://doi.org/10.11998/SeisDmc/SN). The CMT focal mechanism can be obtained via the web page (https://www.globalcmt.org/). The bathymetry/topography data are available from General Bathymetric Chart of the Oceans (www.gebco.net).

## Code availability

The code used for synthesizing waveforms can be accessed at https://data.mendeley.com/datasets/3hvttt7rgy/. All the figures are plotted with Generic Mapping Tools (GMT, https://www.generic-mapping-tools.org/) and further modified with Adobe Illustrator (https://www.adobe.com/products/illustrator.html).

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

## Acknowledgements

We thank Dr. Wenbo Wu (Woods Hole Oceanographic Institution) for his help on waveform modeling approaches. We are grateful to Dr. Zhe Jia

(UCSD) for granting permission to adapt the published image. This research is supported by the Ministry of Education, Singapore, under its MOE Academic Research Fund Tier 3 (Grant number MOE-MOET32021-0002 received by S.W.). This work comprises EOS contribution number 584.

## Author contributions

W.C.: Conceptualization, Methodology, Investigation, Data curation, Formal analysis, Writing – original draft. S.W.: Supervision, Quality Control, Investigation, Writing – review & editing, Funding acquisition. W.W.: Data curation, Writing – review & editing.

## Competing interests

The authors declare no competing Interests.
