## [Peer Review File · Nature Communications]

Subslab ultra low velocity anomaly uncovered by and facilitating the largest deep earthquakeEditorial Note: Parts of this Peer Review File have been redacted as indicated to remove third-party material where no permission to publish could be obtained.

REVIEWER COMMENTS

Reviewer #1 (Remarks to the Author):

Dear authors,

Congratulations on your work. Below you can find my comments to the manuscript, suggestions for improvements and a few questions for you to address, if you will.

The key message of this study is that for events larger than $M_w > 8$ to occur at depths ~ 600 - 660 km, an ultra low velocity anomaly must be present that leads to rupture propagation beyond MOW. Another key message from this study is that high frequency waveform modelling is necessary to reveal the fine structures at the bottom of the transition zone and below subducting slabs, along with a thermodynamic interpretation of seismic tomography results. The small-scale resolution of waveform modelling at the vicinity of the epicentre can unveil complex thermal and compositional structures that cause rupture mechanisms that lead to $M_w 8+$ deep earthquakes.

The authors have made extensive analyses and modelling of the potential hypotheses that can lead to large earthquakes and explain high-frequency P-wave data at such depths. They also tested different factors (anisotropy, attenuation, ratio of P and S velocity, focal mechanism) which can cause similar waveform observations to validate that their results and hypotheses of SULVA is a most likely cause. Their RSTF analysis and finding of in-plane multipathing of the S2 subevent is well supporting their conclusions of multipathing being an indication of the location of slab, small-scale ULV anomaly in the P-wave structure. The authors further use forward modelling to determine the shape, location and amplitude of the ULV anomaly and by template-matching cross-correlation they compare synthetics and observed data to support their inferred model shape and location of this anomaly. The authors explain their methodology and analyses sufficiently well to support their conclusions and results.

This study is significant enough for publication in the journal because it explains one of the not-so-well understood occurrence of large, deep earthquakes. The complexity of structure and composition within the transition zone and the necessity to explain them interdisciplinarily is highlighted by the proposed hypotheses and performed analyses in this paper. Further work on related subject of transition zone thermal and compositional structure at the regions of subduction zones and locations of recorded deep earthquakes are encouraged and motivated by this work. I also find the study important because it shows that waveform modelling and inversion at higher frequencies than usual global and regional tomography studies are needed to better comprehend not only large deep earthquakes but also the slab anomalies that have been proposed by this and a few other studies.

The paper is well presented, although the figures have limited resolution. Perhaps this is a requirement for the initial submission. Otherwise I would recommend that the font-size of all labels become larger as well as the annotations and symbols on maps and plots, and all figures in main and supplementary texts become sharper and have higher resolution.

I suggest that the authors provide a link to a server where anyone can access the repository of the data used for this study (observed waveforms from IRIS and synthetic data) along with a readme file, if possible. The provided links in the "Data availability" section are too generic, directing to a general page of the IRIS DMC.

I would also suggest the authors provide a link to the code used for synthesizing waveforms or if this is not open source, they should perhaps mention the website for information about the code. Additionally, a brief explanation of the numerical waveform modelling code could be useful in the Methods section, so that the readers can understand the advantages and limitations of the modelling.

A brief explanation of the waveform data processing is perhaps missing. The process followed in the paper should somehow be in a way reproducible or at least thoroughly explained.

The approach presented in this paper for synthetic waveform modelling and comparison of observations and models are well justified and supports the results and conclusions in the paper. It would perhaps be useful for completeness, if the authors could comment a bit further on why they have chosen to only focus on the multipathing effects.

Please also explicitly add the reference to the tomographic model used for the cross-sections on Figure 7. The pictorial example is convincing, although it would be useful to include a comment on how the same cross-sections from global 3-D tomographic models derived from different methodologies (eg: from FWI Global adjoint tomography—model glad-m25. *Geophysical Journal International* by Lei, W., Ruan, Y., Bozdağ, E., Peter, D., Lefebvre, M., Komatitsch, D., Tromp, J., Podhorszki, N., Pugmire, D., 2020, <https://doi.org/10.1093/gji/ggaa253>, or from dispersion curves and body wave traveltimes eg: S40RTS: a degree-40 shear-velocity model for the mantle from new Rayleigh wave dispersion, teleseismic traveltime and normal-mode splitting function measurements by J. Ritsema, A. Deuss, H. J. van Heijst, J. H. Woodhouse, *Geophysical Journal International*, Volume 184, Issue 3, March 2011, Pages 1223–1236, <https://doi.org/10.1111/j.1365-246X.2010.04884.x>). How are these models useful to verify their hypothesis?

The manuscript could benefit, in my opinion, if the authors could consider to address or comment on the following points:

- 1) Do the models used for the synthetics account for varying topography along the 660 km discontinuity? How would a more precise topography model be useful to explain the hypothesised SULVA structure?
- 2) Furthermore, it has been shown via synthetic analyses that most of the known topography models may suffer from underestimation of the topographic variation along the 660 km discontinuity (eg Koroni & Trampert, *GJI*, 2016). How would that affect their results regarding the structure's position, extent and resulting buoyancy of/due to SULVA in the case of an elevated 660 km discontinuity which is larger than

10-20 km?

3) Would it be possible to specifically define the "high-frequency" mentioned in the paper (P1 L11, P2 L41, L43, P5 L101, P5 L114)? What would be the suitable frequency range to explain the deployed waveform data, perform modelling of synthetics and be able to properly resolve the proposed SULVA structure?

4) The grid search method deployed to define the relative location and timing has been tried for various combinations. Can the authors please comment on the possibility of overfitting the derived parameters or the effect of the number of combinations on the derived properties used in the grid search?

5) On page 8 lines 161-166, the authors explain the effect of multipathing, providing references to studies that explain this phenomenon as a result of sharp velocity changes. The studies mainly refer to CMB where the velocity changes are indeed dramatic. Are there also studies for more relevant depths and similar abrupt velocity changes within the MTZ (where the pressure and temperature changes are in general smoother than at the CMB)? If yes, they should be cited. Moreover, has the multipathing effect been observed in literature as a consistent effect at MTZ?

6) Could perhaps the authors suggest what other wave or phases would be beneficial to cross-validate their results? Other wave phenomena that could have resulted from slab anomalies and observed in real recordings from large earthquakes and relevant subevents? Any references one could add in the manuscript that can give a better overview for the reader?

I find that the text is well-written and accessible. The authors provide sufficient references to support their results and contextualise their work according to the literature related to slab velocity anomalies, large, deep earthquake investigations and waveform modelling at the subduction zones.

The authors provide most of the references that can support their manuscript. I have made some suggestions for further references which I believe can enhance their work within the context of some questions I have written above.

Reviewer #2 (Remarks to the Author):

Chen et al. explores the initial rupture stage of the Mw 8.3 2013 Sea of Okhotsk deep-focus earthquake using high-frequency teleseismic P waves. The authors successfully identified three subevents during the first six seconds of the rupture, including a subevent (S2) that is roughly 12km shallower than the earthquake hypocenter (S1). Their analysis, supported by data from western US seismic stations, suggested a multi-pathing effect caused by an ultra-low velocity anomaly beneath the slab. From these observations, the authors conclude that this ultra-low velocity anomaly likely contributed to a thermal runaway weakening mechanism, which they propose as the cause of the 2013 Sea of Okhotsk earthquake. While I commend the clever approach of waveform analysis, there are aspects of the findings that might need further substantiation, especially in relation to the thermal runaway mechanisms.

The main concern I have is related to their interpretation of the influence of the detected low-velocity body on the dynamics of the M8.3 earthquake. The authors attribute the process to thermal runaway, a dynamic phenomenon that typically involves significant shear heating (and potentially partial melting) within the "fault-zone" materials, leading to large stress drops and low radiated efficiency. On the other hand, the detected low velocity body would likely have a static effect on the surrounding materials. These two are fundamentally distinct phenomena, and the rough induced-stress estimates provided in the study may not sufficiently validate their claims. Furthermore, it might be relevant to note that thermal runaway often happens at the end stages of rupture, while the study focused on the initial rupture stages, a fact that the study does not seem to address. If the authors aim to establish a connection between the low-velocity body and the facilitation of thermal runaway rupture processes, the inclusion of detailed 3D rupture simulations could be beneficial.

Moreover, I find a potential conflict between the vertical offset of S2 from the hypocenter and the proposed sub-horizontal rupture propagation of the M8.3 earthquake. Could it be possible that the S2 event is a reflection phase from a shallower structure, in line with velocity anomalies? Figure 3c, featuring EU array data, shows a phase arriving about 1s prior to the S2 phase, which could be the mentioned Sx phase, but with significantly larger amplitudes compared to other arrays. The data might benefit from additional exploration and clarification.

Despite these concerns, I appreciate the study's novel approach to waveform analysis. The authors have insightfully identified a phase and endeavored to understand its origin. However, some technical details may require refining. For instance, as noted by the authors, different subevents may have distinct focal mechanisms, which could affect the Green's functions and, in turn, the deconvolution analysis applied in the study. Furthermore, handling the PLD is a delicate task that can easily introduce artifacts, potentially impacting the detection of subtle signals. Thus, I recommend the authors to consider potential uncertainties and biases more comprehensively.

Reviewer #3 (Remarks to the Author):

The manuscript presents a new waveform analysis of the 611-km deep, Mw8.3, Okhotsk earthquake occurring in 2013. The authors make use of dense arrays in China and the US (USArray) to get a view of seismic structure under the slab. They find an ultraslow Vp anomaly (~18%) below the slab that the authors argue contributed to the thermal runaway effect that has been shown to be likely necessary (in previous work) for this event to occur. This is an interesting work with some nice analysis potentially shedding important light on processes of deep earthquake initiation. There are a number of places that reasoning or presentation could be improved which I outline below. With some revisions, this will be a nice article for Nature Communications.

Figure S5

One place that I spent a lot of time scratching my head was in Supplementary figure S5. This seems like an important enough figure to the analysis that it probably should not be relegated to supplementary materials. Still I had a hard time seeing exactly what the authors tell me they are seeing. When I look at the row showing synthetic for S, P, and S+P, I don't see any difference across the three. Same goes for the row S+U, P+U, and U. Is there a mistake here of plotting the same thing multiple times? Or is there some subtle, yet important, difference I should be gleaming from this? The results and interpretation hinge on this, so some additional guidance and explanation is necessary.

Rays are traced in panel (a) for 30, 60, and 90 degree deltas. However, panels (b) and (c) go from 40-80. It would make more sense to trace rays at 40-60-80 to match the synthetics in the following panels. The solid-dashed-solid scheme is also confusing. At first I thought there was a difference between types of rays. Better to make them all the same (e.g., all solid). The caption needs to note the difference between the black and white stars. One has to go to S6 to be sure of their meanings.

How important is this 'plume' anyway? I get that it is there because of a broad low velocity anomaly in the tomography, but the major result is that there is the 'U' (ULVZ) within that. Could 'P' just be a smeared 'U'? I couldn't find anywhere in the paper whether P was helpful or even necessary to interpreting the synthetics.

Odd to me is that the authors mention a V_p anomaly of ~18% being required, but in the supplementary figures, the only 18% anomaly I see modeled is in figure S9 which is a sensitivity analysis and not a comparison with observations. So, why 18%?

Given the vertical difference found between S1 and S2, do the authors view the fault plane as sub-vertical rather than sub-horizontal as previous papers suggest?

Where does the tomography in figure 1b come from? The caption says reference 24, but I don't see it there.

A few details about the SULVA deserve to be fleshed out further. For instance, couldn't the SULVA be due to volatiles rather than melt? It might make more sense for a few 10s of km volume to be high in volatiles than be super hot and melty. And, if we do stick to what's implied by S-U-P, what would the relation of U be to a plume? I think the 'plume' is really a stand-in for the low velocity anomaly and doesn't need to be a plume per se. Either way, though, some explanation is warranted.

I find myself a bit uncomfortable with a 20 km uplift of the 660 immediately adjacent to a downward deflection from the slab. The authors seems forced in that direction to get enough buoyancy out of the SULVA to affect slab stresses in the desired manner, but all the more reason to convince me that an adjacent down-up is feasible. How much dT would that entail and over what spatial extent? It seems rather questionable to keep a few 10s of km volume extra hot and melty next to a slab for a significant amount of time to keep the elevated discontinuity (see above comments on volatiles). This suggests either transience or ongoing replenishment – neither of which the authors mention, let alone address.

In lines 192 & 194, the language needs to be modulated slightly. E.g, change 'clear low-velocity anomaly'

to 'subtle low-velocity anomaly' and 'likely generates the multipathing' to 'potentially generates the multipathing'.

The paragraph starting in line 284 needs some reworking. For instance "To verify if such partial melting is related to the thermal structure... we examine the global tomography ... of deep earthquake". This does not verify whether this is partial melting. It may verify that other deep earthquakes are in proximity to similar velocity anomalies, but doesn't do what the above statement says. I am not sure what is meant by 'clear existent/erosion of low velocity anomaly'. Please clarify what is meant there. In line 294 'entrainment of upper mantle during subduction' is mentioned as a possibility for generating the SULVA. I think this should be 'entrainment of lower mantle during subduction'. Either fix or clarify how the upper mantle could be entrained here.

In figure 1a, enlarge the symbols. The red-blue coloring of E-W in 1a and 1b seems unnecessary. Just noting E-W would be sufficient. In figure 1c, get rid of the topo colors. It is much harder to read than 1d which does not have the topo colored.

At the end of the first section it would help to say what this 'novel' hypothesis is rather than keeping it secret until later.

I hope to see the comments above addressed and see this study in print.

Referee 1

Congratulations on your work. Below you can find my comments to the manuscript, suggestions for improvements and a few questions for you to address, if you will.

The key message of this study is that for events larger than $M_w > 8$ to occur at depths ~ 600 - 660 km, an ultra low velocity anomaly must be present that leads to rupture propagation beyond MOW. Another key message from this study is that high frequency waveform modelling is necessary to reveal the fine structures at the bottom of the transition zone and below subducting slabs, along with a thermodynamic interpretation of seismic tomography results. The small-scale resolution of waveform modelling at the vicinity of the epicentre can unveil complex thermal and compositional structures that cause rupture mechanisms that lead to $M_w 8+$ deep earthquakes.

The authors have made extensive analyses and modelling of the potential hypotheses that can lead to large earthquakes and explain high-frequency P-wave data at such depths. They also tested different factors (anisotropy, attenuation, ratio of P and S velocity, focal mechanism) which can cause similar waveform observations to validate that their results and hypotheses of SULVA is a most likely cause. Their RSTF analysis and finding of in-plane multipathing of the S2 subevent is well supporting their conclusions of multipathing being an indication of the location of subslab, small-scale ULV anomaly in the P-wave structure. The authors further use forward modelling to determine the shape, location and amplitude of the ULV anomaly and by template-matching cross-correlation they compare synthetics and observed data to support their inferred model shape and location of this anomaly. The authors explain their methodology and analyses sufficiently well to support their conclusions and results.

This study is significant enough for publication in the journal because it explains one of the not-so-well understood occurrence of large, deep earthquakes. The complexity of structure and composition within the transition zone and the necessity to explain them interdisciplinarily is highlighted by the proposed hypotheses and performed analyses in this paper. Further work on related subject of transition zone thermal and compositional structure at the regions of subduction zones and locations of recorded deep earthquakes are encouraged and motivated by this work. I also find the study important because it shows that waveform modelling and inversion at higher frequencies than usual global and regional tomography studies are needed to better comprehend not only large deep earthquakes but also the subslab anomalies that have been proposed by this and a few other studies.

A. The paper is well presented, although the figures have limited resolution. Perhaps this is a requirement for the initial submission. Otherwise I would recommend that the font-size of all labels become larger as well as the annotations and symbols on maps and plots, and all figures in main and supplementary texts become sharper and have higher resolution. **Thanks for the suggestion. We have improved the fonts and resolution of the pictures accordingly.**

B. I suggest that the authors provide a link to a server where anyone can access the repository of the data used for this study (observed waveforms from IRIS and synthetic data) along with a readme file, if possible. The provided links in the "Data availability" section are too generic, directing to a general page of the IRIS DMC. **We have stored all the waveform data used in our study at the following link that is included in the revised manuscript: <https://data.mendeley.com/datasets/3hvttt7rgy/1>**

C1. I would also suggest the authors provide a link to the code used for synthesizing waveforms or if this is not open source, they should perhaps mention the website for information about the code.

We have stored related code at the following link and included it in the revised manuscript: <https://data.mendeley.com/datasets/3hvttt7rgy/1> . 2-D finite-difference code is written by Dunzhu Li (https://scholar.google.com/citations?hl=en&user=uRfnOU4AAAAJ&view_op=list_works&sortby=pubdate) and demonstrated in his paper "Global synthetic seismograms using a 2-D finite-difference method"(Li et al., 2014).

C2. Additionally, a brief explanation of the numerical waveform modelling code could be useful in the Methods section, so that the readers can understand the advantages and limitations of the modelling

We added a "2D FD waveform modeling" section in the "Methods" to further elaborate the waveform modelling, as well as the pros and cons of the method.

D. A brief explanation of the waveform data processing is perhaps missing. The process followed in the paper should somehow be in a way reproducible or at least thoroughly explained.

We added a "Waveform data processing" section at the beginning of "Methods" to elaborate more details, both raw data and the processed RSTFs are provided in the data link. Hopefully this is sufficient to allow reproduction of our results.

E. The approach presented in this paper for synthetic waveform modelling and comparison of observations and models are well justified and supports the results and conclusions in the paper. It would perhaps be useful for completeness, if the authors could comment a bit further on why they have chosen to only focus on the multipathing effects.

Thanks for raising this, which is a very good point. In the revised manuscript, as we stated in the "2D FD waveform modelling" section in "Methods" and supplement text S2, the identification of multi-pathing effects is somewhat related to our experiences in high frequency waveform modelling. It is easier to understand that as the modelling frequency increases, the 3D structure (or more complex source) effect will start to appear. This

multipathing feature was later confirmed by sensitivity tests on attenuation, 3D wavefield propagation, anisotropy, variation on $d\ln V_s/d\ln V_p$, as well as on earthquake focal mechanism.

F1. Please also explicitly add the reference to the tomographic model used for the cross-sections on Figure 7.

It was added.

F2. The pictorial example is convincing, although it would be useful to include a comment on how the same cross-sections from global 3-D tomographic models derived from different methodologies (eg: from FWI Global adjoint tomography—model glad-m25. *Geophysical Journal International* by Lei, W., Ruan, Y., Bozdağ, E., Peter, D., Lefebvre, M., Komatitsch, D., Tromp, J., Podhorszki, N., Pugmire, D., 2020, <https://doi.org/10.1093/gji/ggaa253>, or from dispersion curves and body wave traveltimes eg: S40RTS: a degree-40 shear-velocity model for the mantle from new Rayleigh wave dispersion, teleseismic traveltime and normal-mode splitting function measurements by J. Ritsema, A. Deuss, H. J. van Heijst, J. H. Woodhouse, *Geophysical Journal International*, Volume 184, Issue 3, March 2011, Pages 1223–1236, <https://doi.org/10.1111/j.1365-246X.2010.04884.x>). How are these models useful to verify their hypothesis?

We have extended the vertical profiles presented in Figure 7 to examine velocity perturbations in three tomography models: GAP-P4(Fukao & Obayashi, 2013), GLAD-M25(Lei et al., 2020), and S40RTS(Ritsema et al., 2011) (Figure R1). GAP-P4 stands out by revealing the most distinctive and stronger short wavelength features, owing to its calculation of banana-donut kernels at relatively high frequencies (2Hz). It is important to note that the shortest period for 3D synthetics calculated in GLAD-M25 is 17 seconds, and the shortest period for the phase velocity measurements in S40RTS is 40 seconds. Despite their relatively lower resolution, the latter two models also exhibit sub-slab low-velocity anomalies. We have also included this figure as Figure S15 in supplementary materials. More importantly, in contrast with these global tomography models, we also compared the regional travel time tomography model(Conder & Wiens, 2006) at the 2018 Fuji-Tonga M8.2 event. This higher-resolution model reveals a clear low velocity structure with even larger P-wave velocity reduction directly beneath the source of the 2018 M8.2 Fiji earthquake (Figure R2, also as Figure 7d). Also note that this low velocity structure has distorted the geometry of the subducted slab in a way that the velocity anomaly exerts a buoyance beneath the slab.

Figure R1(Figure S15). Vertical profiles depicting wave speed anomalies for three tomographic models: P-wave anomalies for GAP-P4 (top) and GLAD-M25 (middle), and S-wave anomalies for S40RTS (bottom) in the context of the three largest deep-seated seismic events. Black stars enclosed by circles denote the centroid locations of these events.

Figure R2. Left: Tomography images depicting the 2018 Fiji earthquake. The map view of the source region is plotted below, with gCMT focal mechanisms. The star marks the gCMT centroid's location, and the red line represents the surface projection of the tomographic profile. The dashed contours depict the Slab 1.0 model with a depth interval of 100 km. Right: modified from Conder and Wiens, 2006, showing the V_p velocity anomaly from regional tomograms using ocean bottom seismometers (blue circles) and local island stations (blue inverted triangles). The range is indicated by the black box on the left.

The manuscript could benefit, in my opinion, if the authors could consider to address or comment on the following points:

1) Do the models used for the synthetics account for varying topography along the 660 km discontinuity? How would a more precise topography model be useful to explain the hypothesised SULVA structure?

Thanks for raising this comment. We actually did consider various scenarios in our modeling, including the topography of 660-D. After some tests, we found that the modeling result is not sensitive to 660-D topography variation. This is because the velocity perturbation across 660-D is only 4-5%. Such perturbation is deemed inadequate for inducing the observed multipathing effect.

Nonetheless, we conducted additional forward modelling tests, specifically focusing on the topography of 660-D topography (Figure R3). As shown, the topography changes of 660-D only systematically advance or delay the arrival time, does not change the shape of RSTF waveform, further confirm that the multi-pathing effects we observed are from SULVA.

Figure R3. RSTFs obtained under different topographical conditions at 660-D: (a) an elevation of 20 km, (b) a subsidence of 20 km, (c) an elevation with a trapezoid-shaped profile reaching 30 km, and (d) a subsidence with a trapezoid-shaped profile descending 30 km. It is noteworthy that these RSTFs exhibit minimal time shifts and show no substantial waveform distortion. The red star denotes the earthquake source locations, while the thick black lines represent the boundary between the upper and lower mantle.

2) Furthermore, it has been shown via synthetic analyses that most of the known topography models may suffer from underestimation of the topographic variation along the 660 km discontinuity (e.g. Koroni & Trampert, GJI, 2016). How would that affect their results regarding the structure's position, extent and resulting buoyancy of/due to SULVA in the case of an elevated 660 km discontinuity which is larger than 10-20 km?

Like the previous reply, we show that a more exaggerated 660-D topography (Figure R3) will only shift the arrival time in a systematic way, this could be a perfect trade-off with the origin time of S2. Therefore, a more complex 660-D topography will not change the position and the velocity reduction of SULVA. However, it will make a difference in the buoyance calculation, as we also address later.

3-1) Would it be possible to specifically define the "high-frequency" mentioned in the paper (P1 L11, P2 L41, L43, P5 L101, P5 L114)?

They are specified as suggested.

3-2) What would be the suitable frequency range to explain the deployed waveform data, perform modelling of synthetics and be able to properly resolve the proposed SULVA structure?

The appropriate frequency band range we used (up to 0.8 Hz) is determined by balancing the discernibility of subevents and the coherence of the RSTF waveforms. To better illustrate this, we show the RSTFs at several frequency bands (Figure R4, this is also included in the supplement), where waveform complexity gradually increases as the frequency. Our waveform inspection and modelling started from low frequency, we then noticed the multi-pathing effects as we gradually increase the frequency.

Figure R4. RSTFs are generated from diverse bandpass ranges, with the initial 18 seconds prominently emphasized at the uppermost section. The selection of an appropriate frequency range is contingent upon striking a balance between signal coherency and discernibility. Typically, RSTFs exhibit higher coherency when associated with lower frequencies, while they tend to manifest greater distinctiveness when analysing shorter periods.

4)The grid search method deployed to define the relative location and timing has been tried for various combinations. Can the authors please comment on the possibility of overfitting the derived parameters or the effect of the number of combinations on the derived properties used in the grid search?

The model setup (i.e., relative relocation and SULVA) and grid search strategies we employed have been carefully considered and highly simplified, to avoid handling large number of parameters and suppress trade-offs. For instance, in the relative location grid search, the overfitting issue is minimum, as the horizontal offset, vertical offset and origin time different all have different features in the data, therefore the grid search result is robust as these features can be clearly observed in the travel time data. For the radius of SULVA grid search, we realized that the dipping angle of cut1 is not as well-constrained as cut2, as the multipathing effect is mostly affected by the ray paths along cut2, as shown in Figure 5f. The advantage of the full grid search is that we can clearly see the sensitivity and trade-offs (such as those in Figure 5). Our result suggests that the model setup and parameters selection is quite effective in representing the sensitivity of the model parameters.

5)On page 8 lines 161-166, the authors explain the effect of multipathing, providing references to studies that explain this phenomenon as a result of sharp velocity changes. The studies mainly refer to CMB where the velocity changes are indeed dramatic. Are there also studies for more relevant depths and similar abrupt velocity changes within the MTZ (where the pressure and temperature changes are in general smoother than at the CMB)? If yes, they should be cited. Moreover, has the multipathing effect been observed in literature as a consistent effect at MTZ?

Thanks, this is a good point. Besides CMB, several multipathing publications were made for structures within MTZ, including "Upper-mantle structures beneath USArray derived from waveform complexity"(Sun & Helmberger, 2011), and "Imaging subducted slab structure beneath the Sea of Okhotsk with teleseismic waveforms"(Zhan et al., 2014). These references have been duly cited. Notably, these studies also focus on the subduction processes occurring within the MTZ.

6)Could perhaps the authors suggest what other wave or phases would be beneficial to cross-validate their results?

Since SULVA is a very small spatial scale structure with larger amplitude of velocity perturbation, high-frequency waveform is the most possible way to make detection, like in this study, as well as for further validation. The other possible phase could be the P-wave scattering phases at high frequency and the precursors of S-waves.

Other wave phenomena that could have resulted from subslab anomalies and observed in real recordings from large earthquakes and relevant subevents? Any references one could add in the manuscript that can give a better overview for the reader?

One of such phenomena is a shadow zone (or amplitude reduction) imposed by SULVA. This shadow zone could be verified for both large and small events. To our best Knowledge, this is the first report of SULVA. We are in the investigation of other evidence, hopefully we would be able to give more reports and overviews in the future.

Chen et al. explores the initial rupture stage of the Mw 8.3 2013 Sea of Okhotsk deep-focus earthquake using high-frequency teleseismic P waves. The authors successfully identified three subevents during the first six seconds of the rupture, including a subevent (S2) that is roughly 12km shallower than the earthquake hypocenter (S1). Their analysis, supported by data from western US seismic stations, suggested a multi-pathing effect caused by an ultra-low velocity anomaly beneath the slab. From these observations, the authors conclude that this ultra-low velocity anomaly likely contributed to a thermal runaway weakening mechanism, which they propose as the cause of the 2013 Sea of Okhotsk earthquake. While I commend the clever approach of waveform analysis, there are aspects of the findings that might need further substantiation, especially in relation to the thermal runaway mechanisms.

A. Main Concerns.

The main concern I have is related to their interpretation of the influence of the detected low-velocity body on the dynamics of the M8.3 earthquake. The authors attribute the process to thermal runaway, a dynamic phenomenon that typically involves significant shear heating (and potentially partial melting) within the "fault-zone" materials, leading to large stress drops and low radiated efficiency. On the other hand, the detected low velocity body would likely have a static effect on the surrounding materials. These two are fundamentally distinct phenomena, and the rough induced-stress estimates provided in the study may not sufficiently validate their claims. Furthermore, it might be relevant to note that thermal runaway often happens at the end stages of rupture, while the study focused on the initial rupture stages, a fact that the study does not seem to address. If the authors aim to establish a connection between the low-velocity body and the facilitation of thermal runaway rupture processes, the inclusion of detailed 3D rupture simulations could be beneficial.

Indeed, thermal run away mechanism and the SULVA are two fundamentally different phenomena. However, the referee may have misunderstood their relationship that we are proposing. Here we adopt the "dual-mechanism" hypothesis (Zhan, 2020) as the favoured mechanism to explain this largest deep earthquake. This hypothesis encompasses two distinct stages: the initial phase involving metastable olivine transform faulting, followed by a subsequent stage of thermal runaway. The overarching objective of this research is to establish a link between the identified low-velocity structure (unearthed during the investigation of the initial phase) and the occurrence of thermal runaway instability. This connection is established through the consideration of substantial stress conditions, assuming a significant density contrast arising from the velocity perturbation. Such conditions are posited as essential triggers for initiating the thermal runaway process. A 3D dynamic rupture simulation that can fully incorporate low velocity structure and thermal run away for sure would be helpful. However, based on the literature review, thermal runaway mechanism has never been implemented in a 3D dynamic rupture simulation, no mention the inclusion of 3D velocity structure complexity (SULVA) and stress perturbation it produces. Such simulation alone would need an invention of a new set of 3D dynamic simulation code and it deserves an independent investigation, this is beyond our capacity as well as the scope of our study. Here we consider the stress condition needed (e.g., tens to hundreds of MPa stress loading) for thermal run away, as suggested by (Thielmann, 2018), could be met by the buoyance caused by SULVA. Hopefully this finding will encourage further 3D dynamic simulation investigations.

B-1.

Moreover, I find a potential conflict between the vertical offset of S2 from the hypocenter and the proposed sub-horizontal rupture propagation of the M8.3 earthquake. Could it be possible that the S2 event is a reflection phase from a shallower structure, in line with velocity anomalies?

We can rule out S2 to be reflection phase, as it exhibits significantly higher amplitude than S1 in all azimuths (additional Figure S14). It is rare, if not impossible, to produce a reflected phase with amplitude larger than the direct phase. We conduct more synthetic tests to further verify this understanding. We place two shallow reflectors above a seismic source. They are characterized by extreme velocity perturbations (+30% and -30%). We simulated the synthetic waveforms and obtained the RSTFs in the same way as we did for the real data. The RSTF waveforms (Figure R5) show that the amplitude of the reflected phases is notably smaller than the direct phase.

Furthermore, even if S2 were to be attributed to a reflection, it's important to acknowledge that the observed multipathing effect within the North American array necessitates the presence of a distinct low-velocity structure beneath the slab.

Figure R5. RSTFs acquired when incorporating reflectors above the seismic source are presented as follows: (a) with a +30% Vp (P-wave velocity) velocity perturbation and (b) with a -30% Vp velocity perturbation.

B-2. Figure 3c, featuring EU array data, shows a phase arriving about 1s prior to the S2 phase, which could be the mentioned Sx phase, but with significantly larger amplitudes compared to other arrays. The data might benefit from additional exploration and clarification.

This observation could be construed as disparate radiation patterns or focal mechanisms between Sx and EGF (Figure S13 and Figure R6 below), please see more reply below.

C.

For instance, as noted by the authors, different subevents may have distinct focal mechanisms, which could affect the Green's functions and, in turn, the deconvolution analysis applied in the study. Furthermore, handling the PLD is a delicate task that can easily introduce artifacts, potentially impacting the detection of subtle signals. Thus, I recommend the authors to consider potential uncertainties and biases more comprehensively.

The artifacts or large uncertainty introduced by focal mechanism variation of sub-events primarily located at the polarity-flip parts (near the nodal plane) on the beach ball, as illustrated in Figure R6. This is because either the master event or the EGF event's waveform amplitude become very weak, hence the deconvolution ends with either very small or very large RSTF amplitudes due radiation pattern difference. The azimuthal range of the highly unstable RSTFs expands proportionally with the growth of the focal mechanism difference. Nevertheless, when the rotation of the nodal plane direction between the two events is minimal ($<20^\circ$), the RSTFs generally maintain a high level of consistency when it is more than 20° away from the nodal direction. Even when the rotation of focal mechanism is large (e.g., case C in Figure R6), the artefacts due to focal mechanism difference do not show clear azimuthal variation pattern as those observed for Sx (Figure 3e). Therefore, we consider focal mechanism difference might contribute to some uncertainties to our sub-event and SULVA analysis, but the impact is not substantial.

Figure R6. azimuthal profiles of RSTFs generated by the deconvolution at a 50° distance, utilizing an empirical Green's function derived from a focal mechanism characterized by a strike of 12° , a dip of 79° , and a rake of -89° . The individual panels (a), (b), and (c) correspond to different focal mechanisms as follows:

- (a) Focal mechanism with a strike/dip/rake of $2^\circ/79^\circ/-89^\circ$.
- (b) Focal mechanism with a strike/dip/rake of $32^\circ/79^\circ/-89^\circ$.
- (c) Focal mechanism with a strike/dip/rake of $57^\circ/79^\circ/-89^\circ$.

Referee 3

The manuscript presents a new waveform analysis of the 611-km deep, Mw8.3, Okhotsk earthquake occurring in 2013. The authors make use of dense arrays in China and the US (USArray) to get a view of seismic structure under the slab. They find an ultraslow V_p anomaly ($\sim 18\%$) below the slab that the authors argue contributed to the thermal runaway effect that has been shown to be likely necessary (in previous work) for this event to occur. This is an interesting work with some nice analysis potentially shedding important light on processes of deep earthquake initiation. There are a number of places that reasoning or

presentation could be improved which I outline below. With some revisions, this will be a nice article for Nature Communications.

1. Figure S5

One place that I spent a lot of time scratching my head was in Supplementary figure S5. This seems like an important enough figure to the analysis that it probably should not be relegated to supplementary materials. Still I had a hard time seeing exactly what the authors tell me they are seeing. When I look at the row showing synthetic for S, P, and S+P, I don't see any difference across the three. Same goes for the row S+U, P+U, and U. Is there a mistake here of plotting the same thing multiple times? Or is there some subtle, yet important, difference I should be gleaning from this? The results and interpretation hinge on this, so some additional guidance and explanation is necessary.

Rays are traced in panel (a) for 30, 60, and 90 degree deltas. However, panels (b) and (c) go from 40-80. It would make more sense to trace rays at 40-60-80 to match the synthetics in the following panels. The solid-dashed-solid scheme is also confusing. At first I thought there was a difference between types of rays. Better to make them all the same (e.g., all solid). The caption needs to note the difference between the black and white stars. One has to go to S6 to be sure of their meanings. S, P, and S+P, I don't see any difference across the three. Thanks a lot for the suggestion and sorry about the confusion. We have modified the figure accordingly (Figure R7). This figure is to show that slab (S), plume (P) have very limited impact to the RSTF, as shown in the upper row of Figure R7b, only the ultra-low velocity perturbation (U) can produce the multi-pathing effect (lower row in Figure R7b).

Figure R7 (Figure S5). (a) Configuration of the "SUP" 2D model and 1D ray path depicting subevents S1 and S2 at the source region, with the 660 km discontinuity highlighted by dashed lines. The red and blue solid-dashed-solid curves represent the teleseismic P-wave ray paths, originating from the source and extending towards the WA3 and CN arrays, respectively, for distances of 40°, 60°, and 80° as predicted by the PREM earth model. A black star enclosed within a circle denotes subevent S1, whereas a white star represents S2. (b) RSTFs obtained from synthetic waveforms in the direction of the WA3 array using various combinations of "SUP" structures. The labels "S," "U," and "P" correspond to the

slab structure, SULVA, and elliptical plume head structure, respectively. The slab structure exhibits a positive P-wave velocity anomaly with a maximum perturbation of 5% at its center, decreasing linearly towards its edges. Subevents S1 and S2 are located within the slab structure. The SULVA represents a uniform velocity perturbation, while the elliptical plume head structure (labeled as "P") is positioned across the 660 km discontinuity and demonstrates a peak velocity reduction of -3% at the center of the ellipse.

How important is this 'plume' anyway? I get that it is there because of a broad low velocity anomaly in the tomography, but the major result is that there is the 'U' (ULVZ) within that. Could 'P' just be a smeared 'U'? I couldn't find anywhere in the paper whether P was helpful or even necessary to interpreting the synthetics.

Indeed, the "P" part of the model in Figure S5 was added to represent the plume like structure in the tomography model. We did not mention the plume structure in the rest of the paper because such large plume like structure does not make a significant contribution to the RSTF, as shown in Figure S5 (Figure R7). This figure is to illustrate that the emergence of the multipathing effect is mostly due to a relatively compact structure characterized by a substantial velocity reduction (U), as opposed to a more expansive structure featuring a smoother velocity anomaly (P, plume, or S, slab).

2.

Odd to me is that the authors mention a Vp anomaly of ~18% being required, but in the supplementary figures, the only 18% anomaly I see modeled is in figure S9 which is a sensitivity analysis and not a comparison with observations. So, why 18%?

We only use up to 18% Vp reduction in Figure S9 because it is a 3D SEM-DSM simulation, which is much more expensive than the 2D FD simulations. Larger Vp reduction will require even more computational resources, we therefore stopped at 18%. In other 2D sensitivity tests we can go to larger Vp reduction as 2D FD calculation is much more efficient and it is not necessary to be exactly at 18%.

3.

Given the vertical difference found between S1 and S2, do the authors view the fault plane as sub-vertical rather than sub-horizontal as previous papers suggest?

We prefer the initial stage (S1-S2) is sub-vertical rupture due to transformational faulting within metastable olivine wedge, and later rupture as sub-horizontal due to thermal runaway. But note that none of the previous studies have paid attention to the details of this initial rupture.

4.

Where does the tomography in figure 1b come from? The caption says reference 24, but I don't see it there.

Thanks, the citation and reference are added properly.

5.

A few details about the SULVA deserve to be fleshed out further. For instance, couldn't the SULVA be due to volatiles rather than melt? It might make more sense for a few 10s of km volume to be high in volatiles than be super hot and melty.

Indeed, volatiles can also produce large buoyancy, maybe even larger. Besides, from first-principal calculations and lab experiments, it has been demonstrated that buoyancy can be sustained both above and below the 660-D seismic discontinuity by melts enriched with volatiles. Hence, the presence of volatiles is indispensable for elucidating our observations, whether in the form of pure volatiles or volatiles-bearing melts. We have consequently amended this aspect of the interpretation to accommodate this prospect.

And, if we do stick to what's implied by S-U-P, what would the relation of U be to a plume? I think the 'plume' is really a stand-in for the low velocity anomaly and doesn't need to be a plume per se. Either way, though, some explanation is warranted.

Yes 'plume' is a stand-in, but for larger scale with smaller perturbation, a contrast as the SULVA introduced in this study.

6

I find myself a bit uncomfortable with a 20 km uplift of the 660 immediately adjacent to a downward deflection from the slab. The authors seems forced in that direction to get enough buoyancy out of the SULVA to affect slab stresses in the desired manner, but all the more reason to convince me that an adjacent down-up is feasible. How much dT would that entail and over what spatial extent?

Thanks a lot for this comment. If we were to employ a Clapeyron slope of -2MPa/K for the 660-D discontinuity based on Bina and Helffrich's work in 1994 (Bina & Helffrich, 1994), the 20-kilometer uplift would correspond to an approximate temperature change of around 330 Kelvin ($^{\circ}\text{K}$). While temperature anomalies of such magnitude are feasible in the context of plumes, it appears somewhat excessive, particularly considering the relatively limited spatial extent of SULVA. Consequently, we have made adjustments to Figure 7 to reflect a more gradual topography of the 660-D boundary, taking into account the revised 10-kilometer uplift in our revised calculations.

It seems rather questionable to keep a few 10s of km volume extra hot and melty next to a slab for a significant amount of time to keep the elevated discontinuity (see above comments on volatiles). This suggests either transience or ongoing replenishment – neither of which the authors mention, let alone address.

Determining whether the SULVA structure exhibits transience or is subject to ongoing replenishment exceeds the capabilities of the seismological method employed in this study. This is due to the fact that this structure can only be unveiled through the occurrence of a neighbouring significant deep earthquake. Although an analysis of multiple earthquakes in this region may contribute to addressing this question, it is important to note that large deep events are infrequent within a specific geographic area. We will attempt to address this problem in our future work.

7

In lines 192 & 194, the language needs to be modulated slightly. E.g, change 'clear low-velocity anomaly' to 'subtle low-velocity anomaly' and 'likely generates the multipathing' to 'potentially generates the multipathing'.

Thanks for the detailed suggestions, they have been fixed as suggested.

8

The paragraph starting in line 284 needs some reworking. For instance “To verify if such partial melting is related to the thermal structure... we examine the global tomography ... of deep earthquake”. This does not verify whether this is partial melting. It may verify that other deep earthquakes are in proximity to similar velocity anomalies, but doesn't do what the above statement says. I am not sure what is meant by 'clear existent/erosion of low velocity anomaly'. Please clarify what is meant there.

Thank you for your thorough suggestions. We have already revised this section in accordance with your recommendations. The term 'clear existent/erosion of low velocity anomaly' refers to the low-velocity region beneath the slab, which disrupts the integrity of the slab's shape. To enhance the depiction of this characteristic, we have made modifications to Figure 7d by incorporating a regional tomography image from Conder and Wiens 2006 (Conder & Wiens, 2006).

In line 294 'entrainment of upper mantle during subduction' is mentioned as a possibility for generating the SULVA. I think this should be 'entrainment of lower mantle during subduction'. Either fix or clarify how the upper mantle could be entrained here. Low-velocity zones beneath subducting slabs in tomography images are often interpreted as the result of upper mantle entrainment (Tang et al., 2014; Wang et al., 2020). Sub-slab materials are believed to be conveyed by the motion of the descending slabs. For example, these entrained upper mantle materials are posited as the origin of the Changbaishan volcano, as detailed in the study "Changbaishan volcanism in northeast China linked to subduction-induced mantle upwelling." (Tang et al., 2014) We have modified the sentence to make it more clear.

9

In figure 1a, enlarge the symbols. The red-blue coloring of E-W in 1a and 1b seems unnecessary. Just noting E-W would be sufficient. In figure 1c, get rid of the topo colors. It is much harder to read than 1d which does not have the topo colored.

We modified the figure as suggested, reviewer 1 also has a similar comment.

10

At the end of the first section it would help to say what this 'novel' hypothesis is rather than keeping it secret until later.

Thanks for the suggestion, we have now included the hypothesis in the introduction.

References

- Bina, C. R., & Helffrich, G. (1994). *Seismic Discontinuity Topography*. 99.
- Conder, J. A., & Wiens, D. A. (2006). Seismic structure beneath the Tonga arc and Lau back-arc basin determined from joint Vp, Vp/Vs tomography. *Geochemistry, Geophysics, Geosystems*, 7(3). <https://doi.org/10.1029/2005GC001113>
- Fukao, Y., & Obayashi, M. (2013). Subducted slabs stagnant above, penetrating through, and trapped below the 660 km discontinuity. *Journal of Geophysical Research: Solid Earth*, 118(11), 5920–5938. <https://doi.org/10.1002/2013JB010466>
- Lei, W., Ruan, Y., Bozdağ, E., Peter, D., Lefebvre, M., Komatitsch, D., Tromp, J., Hill, J., Podhorszki, N., & Pugmire, D. (2020). Global adjoint tomography - Model GLAD-M25. *Geophysical Journal International*, 223(1), 1–21. <https://doi.org/10.1093/gji/ggaa253>

- Li, D., Helmberger, D., Clayton, R. W., & Sun, D. (2014). Global synthetic seismograms using a 2-D finite-difference method. *Geophysical Journal International*, 197(2), 1166–1183. <https://doi.org/10.1093/gji/ggu050>
- Ritsema, J., Deuss, A., Van Heijst, H. J., & Woodhouse, J. H. (2011). S40RTS: A degree-40 shear-velocity model for the mantle from new Rayleigh wave dispersion, teleseismic traveltimes and normal-mode splitting function measurements. *Geophysical Journal International*, 184(3), 1223–1236. <https://doi.org/10.1111/j.1365-246X.2010.04884.x>
- Sun, D., & Helmberger, D. (2011). Upper-mantle structures beneath USArray derived from waveform complexity. *Geophysical Journal International*, 184(1), 416–438. <https://doi.org/10.1111/j.1365-246X.2010.04847.x>
- Tang, Y., Obayashi, M., Niu, F., Grand, S. P., Chen, Y. J., Kawakatsu, H., Tanaka, S., Ning, J., & Ni, J. F. (2014). Changbaishan volcanism in northeast China linked to subduction-induced mantle upwelling. *Nature Geoscience*, 7(6), 470–475. <https://doi.org/10.1038/ngeo2166>
- Thielmann, M. (2018). Grain size assisted thermal runaway as a nucleation mechanism for continental mantle earthquakes: Impact of complex rheologies. *Tectonophysics*, 746, 611–623. <https://doi.org/10.1016/j.tecto.2017.08.038>
- Wang, X., Chen, Q. F., Niu, F., Wei, S., Ning, J., Li, J., Wang, W., Buchen, J., & Liu, L. (2020). Distinct slab interfaces imaged within the mantle transition zone. *Nature Geoscience*, 13(12), 822–827. <https://doi.org/10.1038/s41561-020-00653-5>
- Zhan, Z. (2020). Mechanisms and Implications of Deep Earthquakes. *Annual Review of Earth and Planetary Sciences*, 48, 147–174. <https://doi.org/10.1146/annurev-earth-053018-060314>
- Zhan, Z., Helmberger, D. V., & Li, D. (2014). Imaging subducted slab structure beneath the Sea of Okhotsk with teleseismic waveforms. In *Physics of the Earth and Planetary Interiors* (Vol. 232, pp. 30–35). <https://doi.org/10.1016/j.pepi.2014.03.008>

REVIEWER COMMENTS

Reviewer #1 (Remarks to the Author):

The authors have addressed my previous comments to a satisfactory degree and made necessary changes in the manuscript and figures.

I have a few more comments that the authors could consider for finalizing their paper submission. Additionally, I suggest the authors improve their explanation and justification of 2-D FD modelling.

Comments:

1. P20 L656: "waveforms"?

2. P20-L661-662: "optimised for GPUs"?

3. The authors should choose to either use a hyphen or not in XD (2D, 3-D, etc.), but make it consistent throughout the text.

4. Perhaps it would make the manuscript more complete if the authors added their analysis of the sensitivity of their hypotheses to the peak-to-peak variation of the 660 discontinuity in the supplementary information along with all relevant references (Figure R3).

5. Regarding authors' answer #6 in the letter: I wonder why these phases mentioned there have not been used as an additional observation to cross-validate their hypotheses or at least the P-wave scattering. Perhaps the frequency range and modelling in this study are sufficient to make some obs-syn comparisons to observe any changes due to SULVA. It seems to me that this would make the case stronger.

6. In the section on 2-D FD modelling, I would suggest the authors comment on their choice of using this code instead of a 3-D more accurate waveform modelling code. Also, I would suggest they add some text about the resolution of the deployed velocity model for the are relative to the maximum resolvable frequency of 1.5Hz. As they have already mentioned a few times, 3-D structure becomes more important with increasing frequency content. Therefore, I think their choice of using a 2-D modelling approach is not fully appropriate. Current computational resources and 3-D codes of wavefield propagation can tackle the simulation of such frequencies (up to 1Hz, which is actually the upper bound of what is used in the paper). Additionally, close to the source (~30 degrees) and for such large magnitudes, the simulation of a line source may be unsuitable. I would like the authors to revise the text with more convincing justifications for their choices. Especially Lines 665-669 are unclear. Please consider revising this part of the paper, for the sake of the advancements and efforts researchers make for more accurate 3-D simulations, which may take more time but nonetheless lead to improved modelling and understanding of data.

Best regards.

Reviewer #2 (Remarks to the Author):

I thank the authors for the thorough revision. I admire the authors for putting forth thought-provoking hypotheses, but perhaps some further clarifications at L56-58 and in the Discussion section may help distinguish the observations and interpretations. If I understand the reply correctly, the authors implicitly assume that the earthquake ruptured via a dual mechanism as proposed in Zhan 2020. Based on the assumption that the mechanism is correct, the authors then propose that the velocity anomaly is the cause of the thermal runaway initiation because it provides a stress perturbation. I agree with the authors that the anomaly must have played a role in controlling the earthquake rupture process. However, the fact that the anomaly may have facilitated the rupture propagation does not directly support the occurrence of thermal runaway. This anomaly-induced thermal runaway argument seems to be built on a few layers of assumptions.

Reviewer #3 (Remarks to the Author):

This is a second review of this paper. In the first round of review I commented that “with some revisions, this will be a nice article for Nature Communications.” The authors have done a commendable good faith job in addressing my comments. Time does not permit me to address revisions to comments from other reviewers, but I have no significant reservations as far as my requested revisions are concerned.

There are some minor issues with the writing that the authors should consider for improvement. For instance, writing in active voice is superior to passive.

Example: “This stress likely stems from...” vs “This stress is likely stemming from...”

There are additional examples I could choose.

I also recommend that the authors look at the paragraph beginning on line 365 and highlight each ‘however’. I think the authors will agree that some rewording would be beneficial.

I look forward to seeing this paper in print.

1 **REVIEWER COMMENTS**

2
3 Reviewer #1 (Remarks to the Author):

4
5 The authors have addressed my previous comments to a satisfactory degree and made
6 necessary changes in the manuscript and figures.

7 I have a few more comments that the authors could consider for finalizing their paper
8 submission. Additionally, I suggest the authors improve their explanation and justification
9 of 2-D FD modelling.

10 **Thanks for the suggestion, we have improved the detailed 2-D FD modelling approach**
11 **description.**

12
13 Comments:

14 1. P20 L656: "waveforms"?

15 2. P20-L661-662: "optimised for GPUs"?

16 **Thank you for the clarification. We have integrated these adjustments into the revised**
17 **description of our 2-D FD modelling approach.**

18
19 3. The authors should choose to either use a hyphen or not in XD (2D, 3-D, etc.), but make
20 it consistent throughout the text.

21 **Thanks for the comment, hyphens have been consistently applied throughout both the**
22 **manuscript and supplement.**

23
24 4. Perhaps it would make the manuscript more complete if the authors added their analysis
25 of the sensitivity of their hypotheses to the peak-to-peak variation of the 660 discontinuity
26 in the supplementary information along with all relevant references (Figure R3).

27 **Thank you for your suggestion. In response, we have repositioned Figure R3 to**
28 **Supplementary Figure 16 and incorporated the corresponding text into the discussion**
29 **section, as per your recommendations.**

30
31 5. Regarding authors' answer #6 in the letter: I wonder why these phases mentioned there
32 have not been used as an additional observation to cross-validate their hypotheses or at
33 least the P-wave scattering. Perhaps the frequency range and modelling in this study are
34 sufficient to make some obs-syn comparisons to observe any changes due to SULVA. It
35 seems to me that this would make the case stronger.

36
37 **When suggested to look into other phases (e.g. P-wave scattering and S-wave precursor)**
38 **in the first reply, we referred to use other smaller events that have simpler rupture**
39 **processes. Because both P-wave scattering phase and S-wave precursor are relatively**
40 **weak, for initial rupture of the Mw8.3 Sea of Okhotsk event, P-wave scattering phase**
41 **overlaps with the direct P-waves from the later ruptures, and the S-wave precursor is**
42 **contaminated by the coda waves from the stronger asperities. Smaller events do not have**
43 **these issues, however, there is no such event nearby, as the Mw8.3 event did not generate**
44 **much aftershocks and the historical seismicity is very sparse. Future efforts on using**

45 smaller events could be made for other SULVA structure detection and validation.
46 Sorry that we did not make it clear in the previous reply.

47
48

49 6. In the section on 2-D FD modelling, I would suggest the authors comment on their choice
50 of using this code instead of a 3-D more accurate waveform modelling code. Also, I would
51 suggest they add some text about the resolution of the deployed velocity model for the are
52 relative to the maximum resolvable frequency of 1.5Hz. As they have already mentioned a
53 few times, 3-D structure becomes more important with increasing frequency content.
54 Therefore, I think their choice of using a 2-D modelling approach is not fully appropriate.
55 Current computational resources and 3-D codes of wavefield propagation can tackle the
56 simulation of such frequencies (up to 1Hz, which is actually the upper bound of what is
57 used in the paper).

58

59 We used 2-D FD rather a 3-D code (e.g. SEM-DSM code as we also used) for most of the
60 synthetic simulations because the predominant multipathing feature in our observations is
61 in-plane, as elaborated in Supplementary Note S2. This characteristic could be adequately
62 modelled with 2-D simulations. The GPU-based 2-D FD code is much more efficient than
63 the 3D SEM-DSM code. For instance, it took ~10 min for a 2-D simulation on a GPU server
64 with three V-100 cards, but it took ~36 hours for a 3D simulation using 64 cores on a cluster.
65 Given the large number of simulations we need to do, using the 2-D FD code is much more
66 feasible.

67 The minimum resolvable spatial scale of the velocity anomaly at ~ 1.5 Hz is comparable to
68 a few wavelength of the used P-waves. Since the P-wave speed is ~ 10 km/s at the depth
69 of 2013 Sea of Okhotsk event, the minimum resolvable scale should be ~ 10 – 20 km.

70

71 Additionally, close to the source (~30 degrees) and for such large magnitudes, the
72 simulation of a line source may be unsuitable. I would like the authors to revise the text
73 with more convincing justifications for their choices. Especially Lines 665-669 are unclear.
74 Please consider revising this part of the paper, for the sake of the advancements and efforts
75 researchers make for more accurate 3-D simulations, which may take more time but
76 nonetheless lead to improved modelling and understanding of data.

77

78 In the 2-D FD code, although a line source is used, a correction from line source to point
79 source is applied to the farfield term (Li et al., 2014). The original writing was referring to
80 the case that the receiver is very close to the source (e.g. < 100 km), in which the nearfield
81 term could not be ignored. In our case, we don't need to concern about the nearfield term,
82 even at the distance of 30 degree.

83 Sorry about this confusion, we have removed this sentence.

84

85

86 Reviewer #2 (Remarks to the Author):

87

88 I thank the authors for the thorough revision. I admire the authors for putting forth thought-

89 provoking hypotheses, but perhaps some further clarifications at L56-58 and in the
90 Discussion section may help distinguish the observations and interpretations.

91 If I understand the reply correctly, the authors implicitly assume that the earthquake
92 ruptured via a dual mechanism as proposed in Zhan 2020. Based on the assumption that
93 the mechanism is correct, the authors then propose that the velocity anomaly is the cause
94 of the thermal runaway initiation because it provides a stress perturbation. I agree with the
95 authors that the anomaly must have played a role in controlling the earthquake rupture
96 process. However, the fact that the anomaly may have facilitated the rupture propagation
97 does not directly support the occurrence of thermal runaway. This anomaly-induced
98 thermal runaway argument seems to be built on a few layers of assumptions.

99

100 Presently, thermal runaway is posited as the sole mechanism for large deep (>300km)
101 earthquakes that rupture beyond the metastable olivine wedge. While the presence of a
102 stress anomaly itself does not guarantee the occurrence of thermal runaway, it stands as
103 a pivotal precondition for thermal runaway. We agree that our study does not provide direct
104 evidence on thermal runaway; rather, we have ascertained that the essential condition for
105 triggering thermal runaway is met, which lends credence to the plausibility of a 'dual
106 mechanism'. We appreciate your comment and have revised the introduction and
107 discussion sections to clarify this point.

108

109 Reviewer #3 (Remarks to the Author):

110

111 This is a second review of this paper. In the first round of review I commented that “with
112 some revisions, this will be a nice article for Nature Communications.” The authors have
113 done a commendable good faith job in addressing my comments. Time does not permit
114 my to address revisions to comments from other reviewers, but I have no significant
115 reservations as far as my requested revisions are concerned.

116

117 There are some minor issues with the writing that the authors should consider for
118 improvement. For instance, writing in active voice is superior to passive.

119 Example: “This stress likely stems from...” vs “This stress is likely stemming from...”

120

121 There are additional examples I could choose.

122

123 I also recommend that the authors look at the paragraph beginning on line 365 and
124 highlight each ‘however’. I think the authors will agree that some rewording would be
125 beneficial.

126 Thank you for your suggestions, these comments have been reflected in the modified
127 version.

128

129 I look forward to seeing this paper in print.

130

131

132 Reference:

133 Li, D., Helmberger, D., Clayton, R. W. & Sun, D. Global synthetic seismograms using a 2-D
134 finite-difference method. *Geophys. J. Int.* **197**, 1166–1183 (2014).
135

REVIEWERS' COMMENTS

Reviewer #1 (Remarks to the Author):

Dear authors

Thanks for revising the manuscript and addressing all comments. From my side there is no new comments.

Best regards.

REVIEWERS' COMMENTS

Reviewer #1 (Remarks to the Author):

Dear authors

Thanks for revising the manuscript and addressing all comments. From my side there is no new comments.

Best regards.

Response:

Dear Reviewer,

Thank you for taking the time to review our manuscript once again and for your positive feedback. We greatly appreciate your thoroughness and diligence in assessing our work. If you have any further questions or require additional information, please do not hesitate to contact us.

Best regards,

Weiwen, Shengji and Weitao